# Estimation of black carbon emissions from Siberian fires using satellite observations of absorption and extinction optical depths

Igor B. Konovalov[1], Daria A. Lvova[1], Matthias Beekmann[2], Hiren Jethva[3,4], Eugene F. Mikhailov[5], Jean-Daniel Paris[6], Boris D. Belan[7], Valerii S. Kozlov[7], Philippe Ciais[6], Meinrat O. Andreae[8,9,10]

[1]Institute of Applied Physics, Russian Academy of Sciences, Nizhniy Novgorod, Russia
[2]LISA/IPSL, Laboratoire Interuniversitaire des Systèmes Atmosphériques, UMR CNRS 7583, Université Paris Est Créteil (UPEC) et Université Paris Diderot (UPD), France
[3]Universities Space Research Association, Columbia, Maryland, USA,
[4]Laboratory of Atmospheric Chemistry and Dynamics, Code 614, NASA Goddard Space Flight Center, Greenbelt, Maryland, USA
[5]Department of Atmospheric Physics, Saint-Petersburg University, St. Petersburg State University, SPbSU, SPbU, 7/9 Universitetskaya nab., 199034, St. Petersburg, Russia
[6]Laboratoire des Sciences du Climat et l'Environnement (LSCE/IPSL), CNRS-CEA-UVSQ, Centre d'Etudes Orme des Merisiers, Gif sur Yvette, France
[7]V. E. Zuev Institute of Atmospheric Optics SB RAS, Tomsk, Russia
[8]Biogeochemistry Department, Max Planck Institute for Chemistry, Mainz, Germany
[9]Scripps Institution of Oceanography, University of California San Diego, La Jolla, CA 92093, USA
[10]Department of Geology and Geophysics, King Saud University, Riyadh, Saudi Arabia

*Correspondence to*: Igor B. Konovalov (konov@appl.sci-nnov.ru)

**Abstract.** Black carbon (BC) emissions from open biomass burning (BB) are known to have a considerable impact on the radiative budget of the atmosphere on global and regional scales, but are poorly constrained in models by atmospheric observations, especially in remote regions. Here, we investigate the feasibility of constraining BC emissions from BB with satellite observations of the aerosol absorption optical depth (AAOD) and the aerosol extinction optical depth (AOD) retrieved from OMI (Ozone monitoring instrument) and MODIS (Moderate Resolution Imaging Spectroradiometer) measurements, respectively. We consider the case of Siberian BB BC emissions, which have a strong potential to impact the Arctic climate system. Using aerosol remote sensing data collected at Siberian sites of the Aerosol Robotic Network (AERONET) along with the results of the Fourth Fire Lab at Missoula Experiment (FLAME-4), we establish an empirical parameterization relating the ratio of the elemental carbon (EC) and organic carbon (OC) contents in BB aerosol to the ratio of AAOD and AOD at the wavelengths of the satellite observations. Applying this parameterization to the BC and OC column amounts simulated with the CHIMERE chemistry transport model, we optimize the parameters of the BB emission model based on MODIS measurements of the fire radiative power (FRP) and obtain top-down optimized estimates of the total monthly BB BC amounts emitted from intense Siberian fires that occurred in May-September 2012. The top-down estimates are compared to the corresponding values obtained using the Global Fire Emissions Database (GFED4) and the Fire Emission Inventory– northern Eurasia (FEI-NE). Our simulations using the optimized BB aerosol emissions are verified against AAOD and AOD data that were withheld from the estimation procedure. The simulations are further evaluated against in situ EC and OC mea-

surements at the Zotino Tall Tower Observatory (ZOTTO) and also against aerosol measurement data collected on board of an aircraft in the framework of the Airborne Extensive Regional Observations (YAK-AEROSIB) experiments. We conclude that our BC and OC emission estimates, considered with their confidence intervals, are consistent with the ensemble of the measurement data analyzed in this study. Siberian fires are found to emit $0.41 \pm 0.14$ Tg of BC over the whole five-months

period considered; this estimate is a factor of 2 larger and a factor of 1.5 smaller compared to that the corresponding estimates based on the GFED4 (0.20 Tg) and FEI-NE (0.61 Tg) data, respectively. Our estimates of monthly BC emissions are also found to be larger than the BC amounts calculated with the GFED4 data and smaller than those calculated with the FEI-NE data for any of the five months. Especially large positive differences of our monthly BC emission estimates with respect to the GFED4 data are found in May and September. This finding indicates that the GFED4 database is likely to strongly

underestimate BC emissions from agricultural burns and grass fires in Siberia. All these differences have important implications for climate change in the Arctic, as it is found that about a quarter of the huge BB BC mass emitted in Siberia during the fire season of 2012 was transported across the polar circle into the Arctic. Overall, the results of our analysis indicate that a combination of the available satellite observations of AAOD and AOD can provide the necessary constraints on BB BC emissions.

## 1 Introduction

Open biomass burning is known to be an important source of black carbon (BC), which is the major absorbing component of carbonaceous aerosol and one of the main atmospheric species contributing to climate forcing (Bond et al., 2013; IPCC, 2013). On the global scale, the radiative forcing of BC, including the effects of BC on ice and snow surfaces, has been estimated to be as high as $+1.1$ W m$^{-2}$ (Bond et al., 2013). In a more recent, observationally constrained analysis, the BC radia-

tive forcing after subtracting the pre-industrial background was estimated to be $+0.53$ W m$^{-2}$ (with the uncertainty bounds of $+0.14$ to $+1.19$ W m$^{-2}$) (Wang et al., 2016), still suggesting that it is quite significant in comparison to the radiative forcing of $1.82 \pm 0.18$ W m$^{-2}$ (Myhre et al., 2013) associated with carbon dioxide (which is the main climate forcer). Open biomass burning (BB) is likely to contribute about 40 % to the total BC emissions (Bond et al., 2013).

As a significant BC source, BB plays an especially important role in climate processes in the Arctic, where the increase of

annual surface temperature in the period since 1875 was almost twice as large as that in the rest of the Northern Hemisphere (Bekryaev et al., 2010). Several studies (e.g., Shindell and Faluvegi, 2009; Flanner, 2013; Sand et al., 2013) indicated that a significant part (up to about 50 %) of this temperature increase could have been induced by BC. There is abundant evidence that BB provides a significant contribution to BC in the Arctic atmosphere in the spring and summer (e.g., Stohl, 2006; Stohl et al., 2006; Warneke et al., 2010; Bian et al., 2013; Hall and Loboda, 2017; Popovicheva et al., 2017; Winiger et al., 2017;

Qi et al., 2017; Xu et al., 2017; Evangeliou et al., 2018). It was also estimated (Evangeliou et al., 2016) that Siberian fires alone contributed almost half (46 %) of the total BC amount deposited in the Arctic over a period of 12 years (2002-2013). Radiative effects associated with BC residing in the Arctic atmosphere include both direct radiation budget changes causing

strong warming of the Arctic surface and significant changes in atmospheric stability and cloud cover (Flanner, 2013; Sand et al., 2013). Significant increases in surface temperature in the Arctic as a result of perturbations of the meridional transport can be caused even by BC residing in the mid-latitude atmosphere (Sand et al., 2013), which indicates that to evaluate correctly the effects of BC on the Arctic climate it is critical to know its concentration not only in the Arctic but also in the atmosphere over adjacent regions such as Siberia. Additionally, deposition of BC on ice and snow has been found to contribute strongly to Arctic warming by decreasing surface albedo and promoting ice/snow melting which, in turn, may result in further surface darkening and provide a positive feedback on the increase of the surface temperature in the Arctic (Hansen and Nazarenko, 2004; Flanner et al., 2009; Flanner, 2013).

The effects of biomass burning on atmospheric composition and climate are commonly evaluated using chemistry transport and climate models relying on data of BB emission inventories, such as, for example, the Global Fire Emissions Database (GFED) (van der Werf et al., 2017), the Global Fire Assimilation System (GFAS) emission data set (Kaiser et al., 2012), the Emissions for Atmospheric Chemistry and Climate Model Intercomparison Project (ACCMIP) inventory (Lamarque et al., 2010), the Fire Inventory from NCAR (FINN) (Wiedinmyer et al., 2011), and the Quick Fire Emissions Dataset (QFED) (Damenov and da Silva, 2015), which are widely used in atmospheric and climate studies. However, emission inventory data are likely to be affected by considerable uncertainties due to a limited knowledge of spatio-temporal characteristics and temperature regimes of fires, as well as due to the lack of reliable estimates of emission factors for a variety of ecosystems and environmental conditions. These uncertainties lead to large discrepancies between emission estimates provided by different inventories. For example, according to the FEI-NE inventory recently developed by Hao et al. (2016), the annual BC emissions from fires in northern Eurasia in the period of 2002-2015 are, on average, a factor of 3.2 larger than those given by the GFED4 (van der Werf et al., 2017) inventory. Using the FEI-NE inventory in the FLEXPART Lagrangian particle dispersion model, Evangeliou et al. (2016) found the model results to be in a reasonable agreement with surface BC concentrations observed at several Arctic stations in the period 2002–2013. On the other hand, a Bayesian inverse modeling analysis based on carbon isotope characterization of BC measurements at Tiksi (East Siberian Arctic) from April 2012 to April 2014 revealed that the best fit of the FLEXPART data to the observations was achieved by reducing the GFED4 fire emissions by 53 % (Winiger et al., 2017); this estimate may, however, reflect uncertainties in the spatial distribution of the GFED4 emissions, as the sensitivity footprints in this particular study cover only a part of Siberia (Winiger et al., 2017). In view of these rather controversial findings and the important role that BC emissions from BB are likely to play in climate processes in the Arctic, it is critical to obtain stronger observational constraints on BC emissions from fires in northern Eurasia and its major BB BC source regions such as Siberia.

Note that the general term "black carbon", which is used throughout this paper, is rather generic and can be broken down into more specific terms, including refractory black carbon (rBC), elemental carbon (EC), and equivalent black carbon (eBC); these terms refer to three major measurement approaches that are used to characterize carbonaceous matter, such as laser-induced incandescence, thermal or thermal optical methods distinguishing between more and less volatile fractions of carbo-

naceous aerosol material, and optical methods based on measurements of light-absorption coefficients (Andreae and Gelencsér, 2006; Bond et al., 2013; Petzold et al., 2013; Lack et al., 2014; Sharma et al., 2017). Accordingly, BC emission data reported by a given emission inventory may be based on one or more specific methods that were employed to evaluate the emission factors used in the inventory. However, a concrete "measure" of BC is usually not specified in BB emission inventories.

The main goal of this study is to investigate the feasibility of constraining the BB BC emissions by using retrievals of aerosol absorption optical depth (AAOD) from satellite measurements performed by the Ozone Monitoring Instrument (OMI) in the near-UV region (Torres et al., 2007; Torres et al., 2013). To achieve this goal, we address the case of the severe fires that occurred in Siberia in 2012 (see, e.g., Konovalov et al., 2014; Antokhin et al., 2018). The other goals of this study are to obtain "top-down" estimates of the monthly BC emissions from fires in Siberia in May-September 2012 and to evaluate the corresponding data of the GFED4 and FEI-NE inventories for this period.

Previous applications of the OMI AAOD retrievals include, in particular, evaluation of BC emissions employed in global aerosol models (Koch et al., 2009; Buchard et al., 2015) and identification of atmospheric variability of AAOD at various scales (Vadrevu et al., 2014; Huang et al., 2015; Eck et al., 2013; Zhang et al., 2017). Zhang et al. (2015) used AAOD retrieved from OMI observations in an inverse modeling analysis involving the GEOS-Chem (Goddard Earth Observing System-chemistry) global model to constrain BC emissions over Southeastern Asia (where the BC emissions are predominantly anthropogenic) for April and October 2006; they found overwhelming enhancements (up to 500 %) in anthropogenic BC emissions in April relative to a priori emission estimates. In this study, the OMI AAOD measurements are analyzed by using simulations performed for the northern Eurasian region (including Siberia) with the CHIMERE chemistry transport model (Mailler et al., 2017).

One of main difficulties in using the OMI AAOD retrievals for constraining BC emissions stems from the well-established fact (see, e.g., Andreae and Gelencsér, 2006; Jethva and Torres, 2011; Bahadur et al., 2012; Mok et al., 2016) that the absorption of UV and shortwave visible radiation by BB aerosol is strongly affected by brown carbon (that is, by the light-absorbing fraction of organic carbon). In view of this fact, explicit modeling of AAOD in the case of BB aerosol as a function of its composition would inevitably involve major uncertainties associated with the assumptions regarding the magnitude of the imaginary part of the refractive index for organic carbon (OC) and the mixing state of aerosol particles; these characteristics are likely strongly variable, depending on sources and atmospheric processing of BB aerosol (Lack et al., 2012, Saleh et al., 2013; Wong et al., 2017; Saturno et al., 2018). To overcome this difficulty, we follow an empirical approach (Konovalov et al., 2017a) that involves parameterization of AAOD as a function of the EC/OC (elemental carbon to organic carbon) ratio and the aerosol extinction optical depth (AOD). This parameterization is based on the analysis of experimental relationships between the single scattering albedo (SSA) of BB aerosol particles and the EC/OC ratio and is fitted to the retrievals of aerosol optical properties from the multi-wavelength measurements made at the Aerosol Robotic Network

(AERONET) sites in Siberia in summer 2012. The relationships between SSA and the EC/OC ratio were reported by Pokhrel et al. (2016) as a result of the Fourth Fire Lab at Missoula Experiment (FLAME-4).

Along with the OMI AAOD retrievals, we use AOD retrievals from satellite measurements made by the Moderate Resolution Imaging Spectroradiometer (MODIS). Numerous studies found the MODIS AOD retrievals to provide useful observational information for evaluation and estimation of BB emissions of aerosol and co-emitted species (e.g., Ichoku and Kaufman, 2005; Matichuk et al., 2008; Kaiser et al., 2012; Petrenko et al., 2012; 2017; Huneeus et al., 2013; Xu et al., 2013; Konovalov et al., 2014; 2015; Reddington et al., 2016). In this study, the MODIS AOD data were used to constrain OC emissions and to optimize the calculated AOD values. Note that since BC is usually a minor component of BB aerosol, AOD is mostly determined by the organic (scattering) fraction of BB aerosol (Reid et al., 2005a); thus estimation of BC emissions using only AOD measurements would require making some assumptions regarding the quantitative aerosol composition. The AAOD measurements are much more sensitive to the BC fraction of aerosol than the AOD measurements, even though the OMI AAOD retrievals are also sensitive to OC due to its strong absorption at shorter wavelengths.

The optimized emissions are validated using independent ground-based and aircraft aerosol measurements performed in different parts of Siberia. Therefore, through the use of a chemistry transport model, this study integrates data from satellite and ground-based remote sensing and in situ and aircraft measurements to not only obtain independent observation-based estimates of BC emissions from Siberian fires but also to ensure their reliability. Moreover, the use of a chemistry transport model allows following the transport path of BB BC emissions in the atmosphere, and to determine the part exported directly to the Arctic.

A general overview of the study design is presented in Fig. 1. The methodology of the study is described in detail in Sect. 2. The results of our analysis are presented in Sect. 3. Finally, Sect. 4 summarizes our findings followed by the concluding remarks.

## 2 Methodology

### 2.1 Observational data

### 2.1.1 AAOD retrievals

We used the OMI AAOD retrievals provided as a part of the OMAERUV (v. 1.8.9.1) Level-2 data product (Torres et al., 2007; 2013) derived by the NASA group from the OMI observations onboard of the EOS-Aura satellite. OMI is a spectrally high-resolution nadir-looking spectrometer measuring the backscattered solar radiance in the ultraviolet and visible regions of the electromagnetic spectrum (Levelt et al., 2006). The OMI measurements provide daily global coverage at a spatial resolution of $13\times24$ km$^2$ at nadir. Aura is a part of NASA's A-train satellite constellation and is in a sun-synchronous ascending polar orbit with a local equator crossing time at 13:45. The OMAERUV algorithm derives the UV Aerosol Index in the 354-388 nm range from radiance observations by making use of the observed departure of the spectral dependence of the

near-UV upwelling radiation at the top of the atmosphere from that of a hypothetical pure molecular atmosphere. Along with the UV Aerosol Index, the OMAERUV data product provides AAOD, AOD and SSA retrieved following a standard look-up-table approach with assumed aerosol models, surface albedo, and aerosol layer height.

The major features of the OMAERUV retrieval algorithm that are relevant in the context of inverse modeling applications of the AAOD data are described in detail by Zhang et al. (2015). Briefly, they are as follows. First, the OMAERUV algorithm identifies one of the three assumed aerosol types, such as BB aerosol, desert dust, and urban/industrial aerosol, representing column aerosol load in each pixel. The selection of aerosol type is based on a scheme that uses coincident and collocated carbon monoxide (CO) observations from AIRS on Aqua and the UV Aerosol Index from OMI (Torres et al., 2013). Second, the algorithm is sensitive to assumptions about the altitude of the aerosol center mass. To address this sensitivity, the AAOD data are retrieved for a set of five different aerosol center mass locations: at the surface and 1.5, 3.0, 6.0 and 10 km above it. The "final" AAOD product derived by OMAERUV is referenced to the monthly climatology aerosol layer height as given by the OMI-CALIOP (Cloud-Aerosol Lidar with Orthogonal Polarization) joint dataset (Torres et al., 2013). Third, the major factor affecting the quality of the aerosol retrievals provided by OMAERUV is sub-pixel cloud contamination. However, compared to AOD and SSA retrievals, AAOD is less affected by cloud contamination due to the partial cancellation of errors in AOD and SSA. OMAERUV reports AOD/SSA/AAOD with the associated quality flags '0' and '1'. While all three retrievals are reliable with the quality flag '0', only AAOD is reliable with either quality flag. Accordingly, the number of reliable AAOD retrievals is greater than the number of those of AOD and SSA.

The OMAERUV product has been validated on a global scale by comparing OMI-retrieved AOD and SSA with the corresponding data derived from ground-based measurements of the sun-sky photometers at the AERONET sites (Ahn et al., 2014; Jethva et al., 2014). The comparison confirmed that the OMI AOD and SSA retrievals are quite reliable. Specifically, the differences between most of the pairs of matched AOD data were found to fall into the expected uncertainty range (the greater of ±30% or ±0.1) for the OMI AOD retrievals for all of the aerosol types, while the majority of collocated SSA retrievals for the "smoke" and "dust" aerosol were found to agree within the expected uncertainties of ± 0.03 in OMI and AERONET inversions. Since AAOD can be expressed through SSA and AOD, these comparisons indicate that the OMI AAOD retrievals are also realistic.

In this study, we made use of the reliable AAOD retrievals corresponding only to the BB type of aerosol. The quality assured values of AAOD at 388 nm for the period from 1 May to 30 September 2012 were projected onto a rectangular 1°×1° grid with hourly temporal resolution. Different values falling into the same grid cell and an hourly period were averaged. We used both the AAOD data sets corresponding to the pre-defined altitudes of the aerosol layer and "final" AAOD product.

**2.1.2 AOD retrievals**

As noted above, the available OMI AAOD retrievals are more abundant than the quality assured retrievals of AOD. For this reason, the OMI AOD retrievals were not used in our analysis. Instead, we employed the Collection 6 retrievals of AOD at

550 nm from the MODIS measurements onboard the Aqua satellite (Levy et al., 2013), which is also a part of NASA's A-train satellite constellation. The MODIS Aqua measurements are typically taken at around the same time as the OMI measurements, as Aqua overpasses the equator daily at the local time of 13:30 in the ascending mode. The merged "Dark Target" and "Deep Blue" AOD retrievals with a nominal horizontal resolution of $10 \times 10$ km$^2$ were obtained for the period from 1

May to 30 September 2012 as a part of the MYD04_L2 data product (Levy et al., 2015). Similar to the AAOD data, the quality assured AOD data for the period from 1 May to 30 September 2012 were projected onto a rectangular 1°×1° grid with hourly temporal resolution and then averaged. The expected uncertainty range of the MODIS AOD retrievals is ±(0.05 + 15 %). A comparison of the Collection 6 MODIS AOD data with the respective AERONET retrievals has shown that this uncertainty range covers the majority (69 %) of the differences between the MODIS and AERONET collocated AOD retrievals

(Levy et al., 2013).

### 2.1.3 Fire radiative power

The Fire Radiative Power (FRP) data (Kaufman et al., 1998; Justice et al., 2002) derived from the MODIS measurements onboard the Aqua and Terra satellites were used in this study to calculate BB emissions of gases and aerosols with the methods developed earlier (Konovalov et al., 2011a; 2014). The FRP data were provided at nominal 1-km spatial and 5-min

temporal resolutions as a part of the MYD14/MOD14 Collection 6 MODIS fire products (Giglio and Justice, 2015a; 2015b; Giglio et al., 2016). The Collection 6 FRP retrieval algorithm makes use of the difference between the 4-μm radiance of a pixel affected by fires with that of a background pixel (Wooster et al., 2012). The Collection 6 Terra MODIS fire products were validated by using reference 30-m fire maps derived from high-resolution Advanced Spaceborne Thermal Emission and Reflection Radiometer (ASTER) images (Giglio et al., 2016). The fire detection omission error was found to be less than

10% for relatively large fires composed of 140 or more fire pixels; however, the ASTER data did not allow quantitative evaluation of the MODIS FRP retrievals which, in general, may be affected by clouds, heavy smoke, or tree crowns.

Our processing of the available FRP data was the same as described in Konovalov et al. (2014). Briefly, we first estimated the FRP density on a 0.2°×0.1° grid covering the northern Eurasian region considered in this study as the ratio of the total FRP in a given grid cell to the observed area of that grid cell. The estimation was done for any Aqua and Terra orbit over-

passing a given grid cell during a given day, and a maximum FRP density value for each grid cell and day in the period from 1 May to 30 September 2012 was selected. Persistent FRP pixels (which may be due to gas flaring) were filtered out. We then estimated the daily mean FRP density by scaling the maximum FRP density with the assumed diurnal cycle of FRP. The diurnal cycle of FRP was estimated using the method proposed by Konovalov et al. (2014) by fitting a Gaussian function approximating the diurnal cycle to the selected FRP daily maxima corresponding to different hours of the day. The daily

mean gridded FRP densities were then used to calculate BB emissions as described below in Sect. 3.2.

### 2.1.4 AERONET data

To characterize the optical properties of BB aerosol in Siberia, we used, along with the satellite retrievals, the aerosol data derived from ground-based measurements of the spectral diffuse sky and direct sun radiation by photometers at the sites of the Aerosol Robotic Network (AERONET) (Holben et al., 1998). Specifically, we used AAOD and SSA retrievals at 440 and 675 nm and AOD observations at 500 and 675 nm. The AAOD and SSA data were obtained from the Version 2, Level-2 (cloud screened and quality assured) aerosol inversion product (Dubovik and King, 2000), while the AOD data were provided from the Version 2, Level-2 direct sun AERONET observations. The uncertainty of the AOD measurements has been estimated to be within ±0.01 in the visible region and ±0.02 at the near-UV wavelengths, and the uncertainty in retrieved SSA has been estimated to be within ±0.03 when AOD at 440 nm is larger than 0.4 (Dubovik et al., 2000). Note that AOD at 440 nm is greater than 0.4 for all retrievals provided as the Level-2 AERONET inversion product (since inversions corresponding to smaller AOD values are considered to be less accurate). Following Konovalov et al. (2017a), here we analyze the AERONET data from sites situated in Siberia. The direct sun measurements and inversions for the fire season of 2012 have been available only from two Siberian sites: Tomsk-22 (56.4° N, 84.7° E) situated in western Siberia and Yakutsk (61.7° N, 129.4 E) situated in eastern Siberia.

### 2.1.5 Measurement data from the Zotino Tall Tower Observatory

To evaluate the simulated concentrations of EC and OC, we used the data collected at the Zotino Tall Tower Observatory (ZOTTO) established at a remote location (about 600 km from the nearest city, Krasnoyarsk) in the boreal forest of central Siberia (Heimann et al., 2014). The geographic coordinates of the observatory are 60.8° N and 89.4° E. Due to the background character of its environment, ZOTTO is suitable for studying natural sources of aerosol and gases in the boreal forest (Chi et al., 2013). The observatory includes a 300 m tall mast that enables probing the atmospheric composition within the planetary boundary layer and capturing the regional concentration signal (Gloor et al., 2001).

Continuous long-term measurements of ambient aerosol carbonaceous fraction (including those of elemental and organic carbon) have been carried out at ZOTTO since 2010 (Mikhailov et al., 2015; 2017). The ambient aerosol was sampled from the top of the tower through a stainless steel inlet pipe and collected on quartz fiber filters. The sampling period varied from 10 h to 480 h, depending on the air pollution level. Concentrations of EC and OC were measured by a thermal–optical carbon analyzer from Sunset Laboratory (OR, USA). The uncertainties of the EC and OC measurements contain a constant part of 0.2 μg[C] cm$^{-2}$ and a multiplicative part of 5 %. Further details on techniques and protocols of the EC and OC measurements at ZOTTO can be found elsewhere (Mikhailov et al., 2017).

### 2.1.6 Aircraft measurements

Aircraft measurements spanning large areas and wide altitude ranges are an indispensable source of the observational data for evaluation of chemistry transport models. Here we use the data collected over Western and Eastern Siberia onboard the

Optik Tu-134 aircraft laboratory in the framework of the Airborne Extensive Regional Observations (YAK-AEROSIB) experiments (Paris et al., 2008; 2009a). In summer 2012, the YAK-AEROSIB measurement campaign was carried out on 31 July and 1 August (Antokhin et al., 2018). On 31 July, the aircraft departed from Novosibirsk (54.9° N, 85.2° E) and arrived in Yakutsk (61.9° N, 128.5° E), with an intermediate landing in Tomsk (56.2° N, 84.7° E). On 1 August, the aircraft departed back from Yakutsk and landed in Novosibirsk. During the flights, the aircraft performed several ascents and descents within the altitude range from about 1 km to 8 km and crossed several major smoke plumes originating from fires in Siberia. Further information on the tracks of the flights that were carried out in the framework of the YAK-AEROSIB campaign in July-August 2012 can be found elsewhere (Antokhin et al., 2018).

In this study, we considered the YAK-AEROSIB observations of mass concentrations of equivalent black carbon (eBC) and fine fraction of aerosol (PM$_{2.5}$) along with the mixing ratio of carbon monoxide (CO). The measurement techniques have been described previously (Paris et al., 2008; 2009a,b). Briefly, eBC was measured with an aethalometer (Panchenko et al., 2000; 2012), which detects diffuse light attenuation by particles collected on a filter. The measurements considered in this study were performed at a wavelength of 640 nm. The sensitivity of the aethalometer is estimated as 0.01 μg m$^{-3}$. The eBC measurements were calibrated against gravimetric measurements of BC mass concentration, using BC particles produced by a pyrolysis generator. It should be noted that the accuracy of aethalometer measurements may be affected by several factors, including the SSA of ambient aerosol particles, particle size, composition and filter loading (see, e.g., Liousse et al., 1993; Sharma et al., 2002; Lack et al., 2014). In particular, the eBC concentration may be strongly overestimated due to greater light scattering by ambient aerosol particles compared to scattering by the soot particles used in the calibration procedure, although this effect can be counterbalanced by lower attenuation for larger filter loadings (see, e.g., Weingartner et al., 2003). Furthermore, the eBC concentration can be different from the EC concentration in the same aerosol sample, simply because eBC and EC measurements represent fundamentally different physical properties of the aerosol (Andreae and Gelencsér, 2006; Bond et al., 2013). Parallel field measurements of eBC (with a standard aethalometer) and EC (with a thermal technique) at an Arctic station (Sharma et al., 2017) revealed that eBC was systematically larger (by about 30 %) than EC in winter and spring (when the aerosol was predominantly anthropogenic) but almost 50 % smaller in summer (when the contribution of BB aerosol might be significant). Although, due to all these reasons, the eBC observations that are widely used for characterizing radiative properties of aerosol in remote regions cannot provide a strong constraint on BC emissions (especially when the BC emissions are interpreted as those of EC), the analysis of the eBC observations performed during the YAK-AEROSIB campaign is useful as it allows us to get an idea about the consistency of different types of BB BC measurements in Siberia.

PM$_{2.5}$ mass concentrations were obtained using the GRIMM 1.109 optical particle counter (GRIMM Aerosol Technik GmbH & Co. KG, Germany) which measures particle number concentration in 31 size channels in the range 0.25–32 μm. Following Burkart et al. (2010), the conversion of number concentration to mass concentration was performed by applying the instrument-specific factor (equal to 1.65) and an additional correction factor, the C-factor, which is dependent on the bulk density

of the sampled aerosol. Burkart et al. (2010) found that the C-factor can be estimated as the ratio of the instrument-specific factor to the aerosol density. Accordingly, assuming that the typical density of BB aerosol is about 1.3 g cm$^{-3}$ (Reid et al., 2005b), we estimated the C-factor for our case to be of 1.27.

Measurements of the CO mixing ratio were made with a modified commercial gas analyzer Thermo 48C (Thermo Environmental Instruments, USA; see Nédélec et al., 2003; Paris et al., 2008). Note that the measurements of the CO mixing ratio were used in this study only for the selection of aerosol measurements representative of BB plumes (as explained below in Sect. 3.4). The eBC, PM$_{2.5}$, and CO observations were matched in time by first averaging the PM$_{2.5}$ concentrations (which were nominally available each second) over a variable period (4 - 16 s) between two consecutive eBC observations and then selecting the closest CO observation (among the data that were nominally available every 4 seconds).

## 2.2 Modeling and analysis

### 2.2.1 Simulations with the CHIMERE model

The simulations considered in this paper were performed with the 2017 version of the CHIMERE model (Mailler et al., 2017), which is an off-line chemistry transport model designed to produce forecasts and simulations of air pollution over a range of spatial scales, from urban to hemispheric. We used the model to simulate mass concentration, composition and optical depth of BB aerosol as well aerosol optical properties (AOD and AAOD) in the absence of fires. Earlier versions of the CHIMERE model have been successfully used in a number of studies of BB aerosol and related atmospheric processes (see, e.g., Hodzic et al., 2007; 2010; Konovalov et al., 2012; Péré et al., 2014; Turquety et al., 2014). The codes of the 2017 version of the CHIMERE model (CHIMERE-2017) have evolved significantly with respect to the codes of the previous versions; the most significant changes are associated with the realization of parallel computations and the representation of the optical effects of aerosols and clouds. However, these changes do not cause major differences in simulations of concentration and composition of BB aerosol. A detailed description of the CHIMERE-2017 model and a list of the recommended model settings (most of which were adopted in our simulations) can be found elsewhere (CHIMERE-2017, 2017; Mailler et al., 2017); hence we describe below only the main features of our computations.

We took into account BB emissions of aerosol and reactive gases along with their other anthropogenic and biogenic sources, including non-BB sources of dust and sea-salt aerosol. The BB emissions were calculated using the MODIS FRP data as explained below in Sect. 2.2.2 The anthropogenic emissions were specified by applying the CHIMERE standard emission interface to the monthly emission data from the global Hemispheric Transport of Air Pollution (HTAP) v2 emission inventory (Janssens-Maenhout at al., 2015). Since HTAP data for the year 2012 were unavailable, we used the corresponding data for the year 2010; the differences between the annual anthropogenic emissions in 2010 and 2012 is unlikely to exceed a few percent in the region considered. Note that the HTAP inventory does not take into account emissions from gas flaring, which, however, was likely to provide only a very minor contribution (less than 2 %) to the BC emissions (Winiger et al.,

2017) in Siberia in summer 2012. Other sources of aerosol and gases were taken into account using the standard CHIMERE procedures described in Mailler et al. (2017).

The aerosol particles of all types were distributed among ten size bins covering the particle diameters from 10 nm to 40 µm following a lognormal size distribution. Based on an empirical BB particle emission model by Reid et al. (2005b), the emis-

sions of BC and OC from fires were distributed among a range of particle sizes using a lognormal particle size distribution with a mass mean diameter of 0.3 µm and a geometric standard deviation of 1.6. A minor fraction of BB emissions, which comprises coarse particles with a typical mean diameter of about 5 µm, was disregarded in our simulations. Note that these coarse particles are not likely to provide a significant contribution to aerosol optical properties in the visible and UV regions of the spectrum (Reid et al. 2005a). The parameters of the distributions for other aerosol types were specified using the stan-

dard settings described in the CHIMERE technical documentation (CHIMERE-2017, 2017).

Aerosol evolution in the atmosphere was simulated with the standard parameterizations implemented in CHIMERE-2017 (Mailler et al., 2017) by taking into account secondary organic aerosol (SOA) formation as well coagulation and dry and wet deposition, except that wet deposition of BB aerosol (which was assumed to be hydrophobic) due to in-cloud scavenging was effectively disregarded by setting the empirical uptake coefficient to be zero. Specifically, the SOA formation was

represented by the scheme proposed and evaluated by Bessagnet et al. (2008). Note that, as shown by Konovalov et al. (2015, 2017b), the atmospheric evolution of aerosol originating from Russian boreal fires may be much more strongly affected by aerosol aging processes involving both primary and secondary semi-volatile organic compounds than represented in the CHIMERE simulations using the standard SOA scheme. Note also that, although experimental findings (Hand et al., 2010) suggest that fresh BB aerosol particles originating from forest fires are composed of predominantly hydrophobic ma-

terial, aerosol aging processes are likely to increase hygroscopicity of BC-containing aerosol particles and to accelerate their removal from the atmosphere by precipitation through in-cloud scavenging (Stier et al., 2006; Oshima et al., 2012; Paramonov et al., 2013). These shortcomings of our simulations are, however, not likely to lead to any significant biases in our estimates of BC emissions, as evidenced by the sensitivity tests discussed in Sects. 3.6.

The chemical evolution of gaseous air pollutants was represented with the MELCHIOR2 chemical mechanism. Following

Konovalov et al. (2017a,b), we introduced two additional trace species that allowed us to estimate the photochemical age of the BB aerosol. One of the tracers ($T_1$) is chemically passive, while the second tracer ($T_2$) reacts with OH (without consuming it) with a constant rate ($k_{OH}$) of $9 \times 10^{-12}$ s$^{-1}$ cm$^3$. This rate constant is chosen to give the reactive tracer a lifetime of about 6 h, given a typical OH concentration in BB plumes of $5 \times 10^6$ cm$^{-3}$ (Akagi et al., 2012). The emissions of both tracers were the same as the BB emissions of OC. The photochemical age ($t_a$) of BB aerosol was evaluated in each grid cell and

hour as follows:

$$t_a = -(k_{OH}[OH])^{-1} \ln([T_2]/[T_1]),$$
(1)

where [T1] and [T2] are column densities of the tracers, and [OH] is the column-average OH concentration within the BB aerosol layer.

As explained in Mailler et al. (2017), the 2017 version of CHIMERE includes the Fast-JX module that computes photolysis rates and some additional diagnostics, including aerosol optical depth. The module first calculates aerosol Mie scattering and absorption for each aerosol species and bin, assuming sphericity of the aerosol particles and using a set of refractive indexes provided with the model. It then computes the radiative transfer in the model atmospheric column and evaluates the actinic fluxes at each model level. Note that the Fast-JX module was slightly modified in this study to enable simulations of AAOD along with AOD. The calculations of AOD and AAOD are performed for five wavelengths (200, 300, 400, 600, and 1000 nm). To evaluate AOD and AAOD at any other wavelength considered in our study, we used a power-law interpolation (assuming a constant Ångström exponent within a given wavelength interval) between the nearest wavelengths from the model's output. As noted in the introduction, the AAOD values computed in the CHIMERE runs taking into account BB emissions were not used in this study because of the high uncertainty of the imaginary part of the refractive index for organic carbon. Instead, we calculated the BB fraction of AAOD by using an empirical parameterization (see Sect. 2.2.3 below) involving the CHIMERE simulations of the BC and OC column amounts and AOD. However, we used AAOD values simulated directly with CHIMERE to characterize the background atmospheric conditions (in the absence of fires).

To enable better consistency between the OMI-derived and simulated AAOD, the simulated vertical profiles of the BB fraction in $PM_{2.5}$ were used to evaluate the altitude of the center of mass of the BB aerosol layer. The OMI-derived AAOD values corresponding to different assumed altitudes of the BB aerosol center of mass were linearly interpolated to the peak height derived from the simulations. The same approach had been employed earlier by Zhang et al. (2015). Taking into account that the aerosol distribution was represented in the AAOD retrieval procedure by a Gaussian profile (Torres et al., 1998), each simulated $PM_{2.5}$ profile was approximated by a Gaussian function and its maximum was considered as the aerosol center of mass.

Following Konovalov et al. (2014; 2017a,b), the simulations were performed in this study on a $1°×1°$ model grid covering a large region in northern Eurasia (35.5° -136.5° E; 38.5° - 75.5° N), including Siberia and parts of Eastern Europe and the Far East. In addition, to simulate the aerosol concentrations at the ZOTTO site, the model was run with a higher resolution of $0.2°×0.1°$ in a nested domain covering a part of central Siberia (86.2°-92.4° E; 57.6°-63.9° N). In the vertical, the model meshes include 12 non-equidistant layers extending from the surface up to the 200 hPa pressure level. The meteorological fields were obtained using the WRF (Weather Research and Forecasting) (version 3.9) model (Skamarock et al., 2008), which was run with a spatial resolution of $50×50$ km$^2$ and driven with the FNL reanalysis data (NCEP, 2017). Boundary and initial conditions were specified using climatological monthly concentrations of aerosols and gases from the LMDZ-INCA chemistry-transport model.

The model runs were done for the period from 18 April to 30 September 2012 both with and without BB emissions of aerosols and gases in the main model domain. Using the results of these model runs, we specified two main modeling scenarios.

The first (base) scenario was assumed to represent the real atmosphere: the modeled data corresponding to this scenario were obtained by including all contributions to aerosol from BB and other sources. The simulations corresponding to the base scenario were performed both with the optimized (as explained in Sect. 2.4.4) and unoptimized BB emissions. The simulation using the optimized emissions is labeled below as "base-opt", and the one using the initial guesses for the parameters involved in our computation of the BB emissions (see Sect. 2.2.2) is labeled as "base-ini". The second scenario was designed to represent aerosol concentrations and optical properties under "background" conditions (in the absence of fires): the data corresponding to this scenario were obtained from a model run (labeled as "bgr") performed without BB emissions in the model domain. The first 13 days (18-30 April) of the runs were considered as the model spin-up period and were excluded from the subsequent analysis.

## 2.2.2 Calculations of BB emissions

Following a number of previous studies (e.g., Sofiev et al., 2009; Konovalov et al., 2011; Kaiser et al., 2012; Huijnen et al., 2016), we calculated BB emissions by assuming a direct instantaneous relationship between the FRP and the emission rate at a time $t$:

$$E^S(t) = \Phi_d \sum_l \alpha \rho_l \beta_l^S h_l(t) F_m^S, \qquad (2)$$

where $E^s$ (g s$^{-1}$ m$^{-2}$) is the emission rate for a species $s$, $\Phi_d$ (W m$^{-2}$) is the daily mean FRP density (see Sect. 2.3), $\alpha$ (g[dry biomass] s$^{-1}$ W$^{-1}$) is the empirical factor relating FRP to the rate of biomass burning, $\rho_l$ is a fraction of a given type, $l$, of the land cover, $\beta_l^s$ (g[model species] g$^{-1}$[dry biomass]) are the emission factors and $h_l(t)$ is the diurnal variation of the FRP density. The relationship (2) also involves the correction factors, $F_m^s$, which are introduced here to enable optimization of the BB emissions for BC and OC in a given month, $m$.

Taking into account the experimental analysis by Wooster et al. (2005), we assumed $\alpha$ to be $3.68 \times 10^{-4}$ g s$^{-1}$ W$^{-1}$. The vegetation land cover fractions, $\rho_l$, were evaluated with the initial resolution of $0.2° \times 0.1°$ using the NCAR USGS land-use dataset (Homer et al., 2004), which was also used to specify the land use data in the CHIMERE model. Consistent with the land use categories defined in CHIMERE, our BB emission model given by Eq. (2) formally takes into account 5 vegetation cover types (needleleaf forest, broadleaf forest, shrubs, grassland, and agricultural land), although no practical distinction was made in this study between BB emissions from needleleaf and broadleaf forest, as well as between BB emissions from shrubs and grassland. The emission factor values for BC and OC (see Table 1) were chosen to be the same as those in the GFED4 inventory; this choice simplifies the comparison of the results of our analysis with the GFED4 data. The emissions of OC were converted into the emission of particulate organic matter (POM) by using a constant OC-to-POM conversion factor (denoted below as $\eta$) of 1.8, which has been found to provide reasonable agreement between the measurements of PM$_{10}$ in central Siberia during the period of major fires in summer 2012 and the mass concentration of the total carbonaceous matter derived from the corresponding EC and OC measurements (Mikhailov et al., 2017; see Fig. 1 therein). The emission

factor values for gaseous species were taken to be the same as in Konovalov et al. (2015, 2017b); they were specified using Andreae and Merlet (2001) and subsequent updates (M.O. Andreae, unpublished data, 2014). The diurnal profile of BB emissions, $h(t)$, was derived directly from the FRP measurements and approximated by Gaussian functions as described in Konovalov et al. (2014, 2015). The correction factors for BC and OC ($F_m^{BC}$ and $F_m^{OC}$) were estimated as described in Sect. 2.2.4 below; their initial guess values (corresponding to the "base-ini" simulation scenario) were equal to unity. The correction factors for other species were set to be equal to 1.3, based on the results of the optimization of CO emissions from Siberian fires in Konovalov et al. (2017b). The fire emissions were first calculated using Eq. (2) with a spatial resolution of 0.2°×0.1° and then projected onto a coarser model grid of 1°×1°.

Figures 2 and 3 illustrate the FRP data used in our analysis and characterize the spatial structure and temporal evolution of the corresponding fires. Specifically, Fig. 2 shows the spatial distributions (within the region covered by the CHIMERE domain) of the daily mean FRP densities, $\Phi_d$, integrated over the study period from 1 May to 30 September 2012 and scaled with the area fraction, $\rho_l$, of a given land cover type; in other words, it shows the integral fire radiation energy per unit area corresponding to a given type of the land cover. The fire radiation energy distributions are shown in Fig. 2 for the two aggregated types of vegetation land cover, one of which includes needleleaf and broadleaf forest and the other comprises all other types of vegetation land cover. Note that the emission estimates are derived in this study for the Siberian region that is depicted in Fig. 2 by red rectangles and referred to below as the study region. These definitions of the study region and model domain allowed us to focus our analysis on the Siberian fires, and at the same time, to take into account the effects of grassland fires in Kazakhstan and large anthropogenic emissions in the European part of Russia. The same study region and model domain had been specified in Konovalov et al. (2014), where, however, BC emissions were not estimated and older versions of the MODIS FRP and AOD data were used. Figure 3 shows the monthly variations of the FRP densities integrated both in time (over a given month) and space (over the study region). Evidently (see Fig. 2), the FRP data are indicative of the major fires that occurred in the study region in 2012 both in western Siberia (north of Tomsk) and in eastern Siberia (east of Yakutsk). The largest fires occurred in the forested areas; the contribution of other (predominantly agricultural and grass) fires to the monthly- and regionally-integrated FRP was comparable to that of the forest fires only in May (see Fig. 3). The fires were, on average, most intensive in July and least intensive in September. The fire radiative energy released from Siberian fires in May, June, and August was nearly the same.

Similar to Konovalov et al. (2014; 2017a,b), the injection height of BB emissions was evaluated using the parameterization proposed by Sofiev et al. (2012) as a function of the observed FRP, the boundary layer height and the Brunt-Väisälä frequency. However, in this study, we used the advanced ("two-step") version of the same parameterization (Sofiev et al., 2012), which allows avoiding underestimation of the heights of BB plumes injected above the atmospheric boundary layer into the free troposphere. Both the boundary layer height and the Brunt-Väisälä frequency were derived from the same WRF output data that were used for the simulations with the CHIMERE model. The BB emissions given by Eq. (2) for each model grid cell were distributed among the model layers proportionally to the weighted number of pixels yielding the injection

height that corresponds to the altitude of a given layer; the weight of each pixel was defined proportionally to the corresponding FRP value. The FRP values in any pixel were assumed to have the same diurnal variation as the BB emissions in Eq. (2).

### 2.2.3 The AAOD parameterization

The method used in this study to evaluate AAOD as a function of the modeled BB aerosol composition and AOD was introduced in Konovalov et al. (2017a). The main assumption underlying our method is that the dependence of the SSA on the ratio of elemental to total carbon, EC/(EC+OC), in the aerosol can be approximated by a linear function. This assumption is based on the results of the Fourth Fire Laboratory at Missoula Experiment (FLAME-4) (Pokhrel et al., 2016), where an almost linear relationship between SSA and EC/(EC+OC) was found for fresh BB aerosol from a wide variety of biomass fu-

els. Furthermore, this assumption has been corroborated by the analysis of aircraft observations of aging BB aerosol (Pokhrel et al., 2016; Konovalov et al., 2017a). Accordingly, based on this assumption, we evaluate EC/(EC+OC) in the atmospheric BB aerosol column using the following empirical relationship:

$$\frac{[EC]}{[EC]+[OC]} \cong (\omega_0^\lambda - b^\lambda)(a^\lambda)^{-1}, \tag{3}$$

where [EC] and [OC] are the column densities of EC and OC, $\omega_0^\lambda$ is the columnar SSA (at wavelength λ) of dry BB aerosol

particles, and $a^\lambda$ and $b^\lambda$ are empirical fitting parameters. Note that $a^\lambda$ is a negative number. Estimates of $a^\lambda$ and $b^\lambda$ for several wavelengths (405, 532, and 660 nm), which are rather close to minus and plus unity, respectively, have been reported by Pokhrel et al. (2016). In particular, $a^{660}$ and $b^{660}$ have been estimated to be −1.11 (±0.04) and 0.99 (±0.004), respectively, using the orthogonal distance regression, ODR, method. These values and Eq. (3) indicate, for instance, that SSA is very close to one (at 660 nm) for pure OC aerosol without absorbing EC.

In this study, we applied Eq. (3) to the AERONET observations at 675 nm. The empirical coefficients $a^{675}$ and $b^{675}$ were evaluated using the power-law extrapolation from the absolute values reported by Pokhrel et al. (2016) at 532 and 660 nm and were found to be −1.12 (±0.04) and 0.99 (±0.004), respectively. Note that using a more sophisticated analysis, Konovalov et al. (2017a) derived EC/(EC+OC) from the AERONET observations at 870 nm. However, we found that the impact of the differences between the EC/(EC+OC) estimates corresponding to the two wavelengths on the empirical parameterization

discussed below was negligible, and so we opted for a simpler and more transparent approach in this study.

The estimates of EC/(EC+OC) derived from the AERONET observations using Eq. (3) were then related to the ratio of AAOD at 388 nm (AAOD[388]) and AOD at 550 nm (AOD[550]). As noted in Sects. 2.1.1 and 2.1.2, we considered the ratio of the AAOD and AOD observations at the different wavelengths rather than direct SSA retrievals because the number of reliable OMI AAOD retrievals is much greater than the number of those of SSA. The AAOD[388] and AOD[550] values were obtained

by extrapolating AAOD and AOD observations at 440 nm to the 388-nm and 550-nm wavelengths, respectively, by using

the corresponding Ångström exponents, which were evaluated using the AERONET observations at pairs of different wavelengths: 440 nm and 675 nm for AAOD and 440 nm and 500 nm for AOD. The relationship between the $AAOD^{388}/AOD^{550}$ ratio and the EC/(EC+OC) ratio was approximated by a linear regression fitted to the data with the ODR method:

$$\frac{AAOD^{388}}{AOD^{550}} = \kappa_1 \frac{[EC]}{[EC] + [OC]} + \kappa_2 , \tag{4}$$

where $\kappa_1$ and $\kappa_2$ are the regression coefficients, which were estimated in Konovalov et al. (2017a) to be 2.05 (±0.86) and 0.014 (±0.028) (the confidence intervals are given in terms of the 90[th] percentile). Note that unlike the standard least-squares method which disregards errors in a predictor variable, the ODR method takes into account random errors in both variables.

In this study, the analysis involving Eqs. (3) and (4) was performed using the same AERONET data (see Sect. 2.1.4) as in Konovalov et al. (2017a) but with relaxed selection criteria. Specifically, instead of requiring that AOD at 500 nm should exceed the fixed value of 0.5, we demanded that $AOD^{550}$ derived from the AERONET measurements should be at least a factor of two larger than the corresponding "background" $AOD^{550}$ values predicted by CHIMERE (see Sect. 2.2.1). We also did not put any restrictions on the photochemical age of BB aerosol [whereas Konovalov et al. (2017a) required that the photochemical age must not exceed 30 h]. While the restriction on the photochemical age allows diminishing the risk that the relationship between SSA and the EC/(EC+OC) can be affected by morphological changes in aged BB aerosol particles, it also strongly reduces the amount of data available for the analysis and greatly increases the statistical uncertainty of the regression coefficients $\kappa_1$ and $\kappa_2$. We assume that the effects of aging processes are manifested as deviations of the data points from the linear relationship given by Eq. (4), and thus can be taken into account in the confidence intervals of the optimal estimates of $\kappa_1$ and $\kappa_2$. Furthermore, unlike Konovalov et al. (2017a), we did not assume that the selected SSA observations are fully representative of BB aerosol (in other words, we did not assume that the impact of the background fraction of aerosol on the observed SSA can be disregarded). Instead, we derived SSA for BB aerosol particles from the AERONET retrievals of AAOD and AOD by using the background AAOD and AOD values predicted in the CHIMERE simulation without BB emissions. That is, we evaluated $\omega_0^{\lambda}$ as follows:

$$\omega_0^{\lambda} \cong 1 - \frac{AAOD_o^{\lambda} - AAOD_b^{\lambda}}{AOD_o^{\lambda} - AOD_b^{\lambda}} , \tag{5}$$

where the subscripts "o" and "b" denote the observations and the simulations for the "bgr" scenario (without BB emissions), respectively.

As the validity of Eq. (3) has been demonstrated only for dry aerosol, any AERONET observations corresponding to average relative humidity in the aerosol column (RHC) greater than 60% were disregarded [similar to Konovalov et al. (2017a)]. The values of RHC were derived from the CHIMERE simulations.

Figure 4 demonstrates the linear relationship between the $AAOD^{388}/AOD^{550}$ and the EC/(EC+OC) ratios (see Eq. 4) that was obtained in this study. In spite of a considerable scatter of the data points, the relationship is rather well constrained because of the large number (equal 66) of selected observations. The regression coefficients, $\kappa_1$ and $\kappa_2$, are estimated to be 2.31 ($\pm 0.24$) and 0.012 ($\pm 0.008$), respectively. Note that a non–zero intercept ($\kappa_2$) is indicative of a contribution of brown carbon

to the imaginary part of the BB aerosol refractive index, but it should also be noted that the brown carbon content in aerosol particles can, in principle, correlate or anti-correlate with the BC content. The confidence intervals were evaluated using the bootstrapping method in terms of the 68.3 percentile (1-sigma) and take into account both random uncertainties in the EC/(EC+OC) and $AAOD^{388}/AOD^{550}$ ratios and the uncertainty in the empirical coefficients $a^{675}$ and $b^{675}$. It is noteworthy that, taking into account the uncertainty ranges, these estimates are entirely consistent with the corresponding estimates (see

above) obtained earlier (Konovalov et al., 2017a) with a much smaller number (20) of selected data points.

To evaluate the impact of the possible biases in the simulated data involved in Eq. (4) on our estimates of $\kappa_1$ and $\kappa_2$, we performed several sensitivity tests (see the Supplementary material, Sect. S1), in which $AAOD_b^{\lambda}$ and $AOD_b^{\lambda}$ were scaled with constant factors. The test results indicate (see Figs. S1-S3) that our optimal estimates of the regression coefficients $\kappa_1$ and $\kappa_2$ are sufficiently robust with respect to possible biases in $AAOD_b^{\lambda}$ and $AOD_b^{\lambda}$.

Based on the empirical parameterization given by Eq. (4), we could predict $AAOD^{388}$ for the BB aerosol in a given model grid cell using the model output data as follows:

$$AAOD^{388} \cong AOD^{550}\left[\kappa_1 \frac{[BC]}{[BC]+[POM]\eta^{-1}} + \kappa_2\right], \tag{6}$$

where [BC] and [POM] are the BC and POM column densities that were simulated together with $AOD^{550}$ with the CHIMERE model by taking into account only BB emissions of aerosol (see Sect. 2.2.1), and $\eta$ is the OC-to-POM conversion

factor of 1.8 (see Sect. 2.2.2).

### 2.2.4 Optimization procedure

We inferred optimal BB emissions of BC and OC by following an inverse modeling approach (Enting et al., 2002) which, in a general case, suggests that the emissions specified in an atmospheric model can be constrained by analyzing the differences between observations of the atmospheric composition and corresponding simulations. Our inverse modeling analysis was

aimed at optimizing the correction factors, $F_m^{BC}$ and $F_m^{OC}$ (see Eq. 2), for the emissions of BC and OC in each month, $m$, of the study period (May-September 2012). Accordingly, the monthly values of the correction factors constitute the components of the state vectors, $\mathbf{F}^{BC}$ and $\mathbf{F}^{OC}$ of our inverse modeling problem. The same value of a correction factor for a given species and a given month applies to each grid cell in the model domain. We require that the optimal estimates of $\mathbf{F}^{BC}$ and $\mathbf{F}^{OC}$ enable eliminating the relative differences between the mean values of both AOD and AAOD simulated with optimized

BB emissions and their pre-selected matchups derived from satellite measurements for a given month:

$$\frac{\left|\left\langle V_{co}\right\rangle - \left\langle V_{cs}\right\rangle\right|}{\left\langle V_{co}\right\rangle} < o; \tag{7}$$

where $V_{co}$ are the daily AOD (when $c$ equals 1) or AAOD (when $c$ equals 2) values derived from satellite observations, $V_{cs}$ are the simulated counterpart of $V_{co}$, the angular brackets denote averaging over the available data (pre-selected as explained below) for the study region in a given month, and $o$ is an arbitrarily small number, which, for definiteness, was set to be of

$3\times10^{-2}$ in this study both when $c$ equals 1 and 2. Note that the value of $o$ approximately characterizes the relative numerical error (imprecision) of the correction factor estimates. The values of $V_{cs}$ are dependent on $\boldsymbol{F^{BC}}$ and $\boldsymbol{F^{OC}}$, which were optimized independently for each month by assuming that the simulated values of AAOD and AOD in a given month are independent of the BB emissions in any other months. To better isolate different months in the estimation procedure, we established a "buffer" between each two neighboring months, comprising five days that were excluded from the analysis. Note

that BB BC and OC transported into the study region from outside of the model domain are regarded as a part of background concentrations of these species.

The optimization of $\boldsymbol{F^{BC}}$ and $\boldsymbol{F^{OC}}$ in accordance with Eq. (7) is equivalent to establishing a simple balance between spatially- and temporally-averaged AAOD (or AOD) retrievals and their simulated matchups on a monthly basis. Note that the criterion given by Eq. (7) would not be sufficient if we were interested not only in constraining total monthly emissions but also in improving their spatial structure. A more general approach to the estimation of BB emissions using satellite observations

involves minimization of the least-square differences between the observations and simulations (see, e.g., Konovalov et al., 2014; 2016; Heymann et al., 2017). However, it was shown (Konovalov et al., 2011b) that the application of the least-square method may result in an underestimation of BB emissions in the presence of multiplicative model errors (which may be due to random uncertainties in the spatial structure and temporal evolution of BB emissions); consistently with the analysis by

Konovalov et al. (2011b), some negative biases (10-15%) in simulated AOD values were found in Konovalov et al. (2014; 2017a) after optimization of BB emissions in Siberia. Such biases are "automatically" avoided in the optimization method used in this study. Note also that improving the spatial structure of the emissions would require much larger computational resources than could be used for this study or more sophisticated computational tools, such as an adjoint model, which was not available for CHIMERE-2017. Furthermore, the findings from a previous study of BB emissions in Siberia (Konovalov

et al., 2014) indicate that increasing the dimension of the state vector would result in a very large uncertainty of the optimal estimates of its components, at least when no a priori constraints (in the Bayesian sense) and explicit quantitative assumptions about the magnitudes of model and measurement errors are used. Fixing the spatial structure of the emissions, on the other hand, unavoidably results in an aggregation error of the top-down emission estimates (Kaminski et al., 2001), as the actual emission fields can be substantially different from those specified in the simulations due to crude representation of

spatial and temporal variability of the factors involved in our emission model (see Eq. 2) as well as due to uncertainties in the FRP observations. However, the aggregation error is unlikely to be considerable in our case, as the satellite observations

are expected to be representative of all areas in the study region where BB emissions were important, and so the contributions of random errors in the emission values for different grid cells to the uncertainty of our total monthly emission estimates are likely to compensate each other. Possible uncertainties associated with the optimization criterion given by Eq. (7) are discussed in more detail in Sect. S2 (see the Supplementary material).

The data selected for our optimization procedure satisfied the following two criteria. First, taking into account the limitations of the empirical parameterization given by Eq. (4), we disregarded any data points (on an hourly basis) corresponding to RHC (see Sect. 2.2.3) greater than 60%. The remaining hourly data matching the corresponding hourly data from the satellite observations were averaged on a daily basis. Note that we did not require the AAOD observations to overlap with the AOD observations in space and time, as the estimates of $F^{BC}$ and $F^{OC}$ are supposed to be representative of BB emissions in the

whole study region. Second, we tried to ensure that the selected data contained a sufficiently strong "signal" from BB aerosol, so that the emission estimates would not be strongly affected by possible biases in the simulated background AOD or AAOD values. Specifically, we required:

$$(AOD - AOD_b)AOD_b^{-1} > \gamma \ ,$$   (8)

where $AOD$ includes both the background and BB components, $AOD_b$ represents the background component of AOD, and $\gamma$

is a constant. This criterion, which is aimed at removing any data points for which the contribution of fire emissions to AOD is small, was applied in each grid cell to the daily AOD data from both observations and simulations. As a base case option, we put $\gamma$ equal to unity; other values were considered in sensitivity tests described in Sect. 3.6. To enable validation of our simulations against an independent subset of the OMI-derived AAOD and MODIS-derived AOD data, one third of the daily data points satisfying the above criteria were randomly withheld from the estimation procedure to constitute a validation

subset of the satellite data.

The optimization problem defined by Eq. (7) was solved iteratively. In each iteration $i$, $\mathbf{V_{cs}}$ was computed using the corresponding estimates of the correction factors, $\mathbf{F}^{OC}(i)$ and $\mathbf{F}^{BC}(i)$, and the improved estimates, $\mathbf{F}^{OC}(i+1)$ and $\mathbf{F}^{BC}(i+1)$, were obtained as follows:

$$\mathbf{F}^{OC}(i+1) = \frac{\langle V_{1o} - V_{1sb} \rangle}{\langle V_{1s} - V_{1sb} \rangle} \mathbf{F}^{OC}(i) \ ,$$   (9)

$$\mathbf{F}^{BC}(i+1) = \frac{\langle V_{2o} - V_{2sb} \rangle}{\langle V_{2s} - V_{2sb} \rangle} \mathbf{F}^{BC}(i) \ ,$$   (10)

where $V_{1s}$ and $V_{2s}$ are calculated using the values of $\mathbf{F}^{OC}(i)$ and $\mathbf{F}^{BC}(i)$, and $V_{1sb}$ and $V_{2sb}$ are the simulated background values of AOD and AAOD. Note that the optimization problem considered is not strictly linear, particularly because the results of the application of the second criterion (see Eq. 8) to the AOD simulated with different OC emissions can be different. None-

theless, the nonlinearities are relatively weak, and the convergence of the simple iteration procedure given by Eqs. (9) and (10) is ensured as long as the BC contribution to AOD is small compared to that of OC. In this study, the initial guesses for $F^{OC}$ and $F^{BC}$ (corresponding to the "base-ini" case) were equal to unity, and the convergence criterion given by Eq. (7) was satisfied after four iterations.

The uncertainty in the optimal estimates of $F^{OC}$ and $F^{BC}$ was evaluated by means of a bootstrapping technique (Efron, 1993) using Eqs. (9) and (10) which were applied to the optimized estimates of the correction factors and corresponding simulations. Specifically, the AOD and AAOD data involved in Eq. (9) and (10) were randomly sampled (with replacements) 3000 times, and the spread of the correction factor values from the left-hand part of Eqs. (9) and (10) was used to evaluate the confidence intervals for the optimal estimates of $F^{BC}$ and $F^{OC}$. To take into account possible spatial and temporal covariances of
the model and/or measurement errors, we ensured that the size of any sampled dataset is not larger than the number of the available (in the optimization subset) data points, for which the distances between them (both in space and time) are larger than the corresponding de-correlation length/time scales (which were evaluated separately both for AAOD and AOD). As a result of this limitation, the size of any sampled monthly dataset was several times smaller than the size of the original optimization dataset for a given month. In each iteration of the bootstrapping procedure, we randomly changed the parameters of
the AAOD parameterization given by Eq. (6) by sampling them from a Gaussian distribution with the standard deviations evaluated above (see Sect. 2.2.3).

Furthermore, we took into account that the AOD simulated by CHIMERE may be biased and/or not sufficiently representative of the variability in the mass extinction efficiency ($\alpha_e$) of the actual BB aerosol in Siberia. Based on BB aerosol properties summarized by Reid et al. (2005a), we assumed that the regional scale variability of $\alpha_e$ for BB aerosol can be characte-
rized by means of the confidence intervals (in terms of the 90[th] percentile) of 0.7 m$^2$ g$^{-1}$. Accordingly, we took into account the uncertainty of $\alpha_e$ by sampling its value in each iteration of the bootstrapping procedure from a Gaussian distribution with a standard deviation of 0.43 m$^2$ g$^{-1}$. Note that the average value of $\alpha_e$ for BB aerosol in our simulations is found to be of 4.92 m$^2$ g$^{-1}$, which is rather close to the likely value of 4.7 m$^2$ g$^{-1}$ suggested by Reid et al. (2005a).

Finally, we took into account that the estimates of both $F^{BC}$ and $F^{OC}$ could be affected by the uncertainty of the assumed
value of the OC-to-POM conversion factor ($\eta$) (see Sect. 2.2.2 and Eq. (6) in Sect. 2.2.3): larger values of $\eta$ would yield smaller values of the correction factors. The BB aerosol composition measurements performed in different regions of the world and summarized by Reid et al. (2005b) suggest that the POM/OC ratio is likely to range from 1.4 to 1.8. On the other hand, Turpin and Lim (2001) indicated that the POM/OC ratio in non-urban aged aerosol affected by wood smoke could be as large as 2.6. However, we believe that the reported extreme values of the POM/OC ratio do not characterize the range of
the uncertainty of the assumed value of $\eta$ (equal 1.8), as this estimate is supposed to represent the average properties of BB aerosol of different origin and age in the vast region considered in this study. For definiteness, we characterized the uncertainty of $\eta$ by means of a Gaussian distribution with a standard deviation of 0.2. This value corresponds to an uncertainty range of $\eta$ from about 1.5 to 2.1 in terms of the 90[th] percentile confidence intervals.

Note that the mean POM/OC ratio representative of the ensemble of aerosol observations considered in this study can actual-ly be different from that representative of BB aerosol emissions, in contrast to our assumption that η has the same value both for the observed aerosol and the fresh emissions. The variability of the POM/OC ratio in ambient (aging) aerosol may partly be due to the formation of SOA from the oxidation of semi-volatile organic compounds (SVOCs) and other processes in-volving SVOCs. These processes, which have been shown to significantly affect BB aerosol evolution in Siberia and to have a potential to cause strong biases in OC emission estimates inferred from AOD measurements (Konovalov et al., 2017b), are not taken into account in the simulations performed in this study. To get some idea about the impact of this omission on our BC emission estimates, it is useful to transform Eq. (6) by taking into account that (i) POM constitutes the major component of BB aerosol, typically accounting for about 80% of its mass, and (ii) the BC mass fraction is about 10 times smaller than that of OC in BB aerosol in temperate/boreal forest (Reid et al., 2005b). Accordingly, by assuming that AOD is determined only by POM and disregarding the contribution of BC to total carbon concentration, Eq. (6) can be approximated as follows:

$$AAOD^{388} \approx \alpha_e(\kappa_1\eta[BC] + \kappa_2[POM]) , \tag{11}$$

where $\alpha_e$ is the mass extinction efficiency discussed above. According to Eq. (11), AAOD may be underestimated in our simulations in the cases with fresh aerosol (where η and [POM] are likely underestimated) but overestimated in the cases with aged aerosol. However, it seems reasonable to expect that such biases in AAOD cannot cause a significant bias in our optimal estimates of $F^{BC}$ and BC emissions as long as the assumed value of the factor η is representative of the average value of this factor over the ensemble of the all (both fresh and aged) plumes observed from the satellites and as long as the simu-lated AOD values (and, accordingly, POM columns) are, on average, consistent with the AOD observations. Furthermore, taking into account that the term proportional to $\kappa_2$ in Eq. (11) is typically much smaller than that proportional to $\kappa_1$ (see Fig. 4), the possible biases in AAOD due to BB aerosol aging are effectively included in our confidence intervals for BC emis-sion estimates by considering the uncertainties in η and $\alpha_e$ as noted above. Any uncertainty of our estimates of $F^{BC}$ and BC emissions due to model errors in the spatial and temporal distributions of the POM columns and AOD is also taken into ac-count in the respective confidence intervals as explained above. The robustness of our BC emission estimates with respect to treatment of SOA formation processes in our model is confirmed by a sensitivity test reported in Sect. 3.6.

## 3 Results

### 3.1 Optimal estimates of the correction factors for BB emissions of BC and OC

The optimal estimates of the correction factors, $F^{BC}$ and $F^{OC}$, and their uncertainties for each of the five months considered are reported in Table 2; note that the subscript *"m"* is omitted here and below for brevity. The estimates range from about 2.3 (in May) to 3.7 (in September) for BC and from 1.5 (in May) to 2.7 (in August) for OC. Table 2 also lists our estimates of the $F^{BC}/F^{OC}$ ratios, which range from about 0.7-0.8 (in June - August) to 1.5-1.7 (in May and September). The estimates of both $F^{BC}$ and $F^{OC}$ as well as of their ratios are reasonably well constrained by the observations: the respective uncertainties

are less than or about 35% for $F^{BC}$ and less than 30% for $F^{OC}$ in the summer months. The uncertainties are largest in the estimates of $F^{BC}$ and $F^{OC}$ for September (45% and 47%, respectively). This is not surprising taking into account that the fires were relatively small in that month (see Fig. 3), and so the number of the available observation data points in the optimization dataset is many times lower for September (58) than, e.g., for July (3017).

The monthly variations in both the correction factors and their ratios exhibit a rather pronounced seasonal pattern. Specifically, the values of $F^{OC}$ are smaller in May and September than in the summer months. In contrast, the values of $F^{BC}$ and the $F^{BC}/F^{OC}$ ratio are much smaller in the summer months than in May and September. Although the differences between the correction factors for different months are mostly not statistically significant, a more than two-fold decrease in the $F^{BC}/F^{OC}$ ratio in June and July is statistically significant with respect to both May and September.

While the monthly variations of $F^{BC}$ and $F^{OC}$ may, in principle, account for changes in both the emission factors and in the conversion factor $\alpha$ (see Eq. 2), the variations in the $F^{BC}/F^{OC}$ ratio may be explained by changes only in the ratio of the BC and OC emission factors. It seems possible that the variations of the $F^{BC}/F^{OC}$ ratio are partly associated with high variability of the contributions of different fire types (featuring different emission factors) to the observed FRP (see Fig. 3): specifically, the contributions of agricultural and grass fires to the integral FRP were 41 and 21 % in May and September, respectively,

but only 14, 10, and 9 % in June, July and August. To examine this possibility, we performed an additional estimation (see the Supplementary material, Sect. S3) using the AAOD and AOD observations only over a selected sub-region (50-57°N, 60-115° E, see Fig. 2) where the relative contribution of agricultural and grass fires to FRP was much larger and more uniformly distributed across the different months than in the whole region (see Fig. S4). The estimates of the $F^{BC}/F^{OC}$ ratio obtained for this sub-region (see Table S1) show much smaller (and statistically insignificant) variations between different

months, as compared to the corresponding estimates for the whole region (in particular, the $F^{BC}/F^{OC}$ ratio has decreased in May but increased in the summer months), whereas monthly variations of the $F^{BC}$ and $F^{OC}$ factors themselves (see also Table S1) have even increased. Therefore, this additional analysis supports the possibility that the monthly variations of the $F^{BC}/F^{OC}$ ratio are associated with different fire types and, therefore, indicates that that the emission factors (for BC or OC or the both species) specified in the GFED4 inventory and in our simulation are biased in case of at least one fire type. However-

er, it should be noted that the variability of the $F^{BC}/F^{OC}$ ratio for the whole study region can also be explained by other reasons, such as spatial variability of the emission factors across different ecosystems in the region considered, as well as by the emission factor monthly variability which is not represented by the constant emission factor values specified in the GFED4 inventory (see Table 1). Based on the limited amount of the available data, we cannot exclude these alternative explanations.

**3.2 Evaluation of the optimized simulations of AAOD and AOD**

In this section we examine whether the simulations that have been employed in the inverse modeling analysis are sufficiently reasonable and representative of the observations that have not been used for optimization of the BB emission parameters. To this end, we compare our simulations, in which the BB emissions have been computed using the correction factors pre-

sented above, with a validation subset of the satellite data (see Sect. 2.2.4). A comparison of our simulations with in situ measurements is presented in the subsequent two sections.

Figure 5 presents the spatial distributions of the temporally averaged AAOD and AOD values over the study region according to the satellite observations and our simulations performed both with the optimized BB emission and with zero BB emissions. Note that blank pixels indicate that either the satellite observations are available for less than two days in these grid cells, or the observed and/or simulated data have not been included in the validation subset according to the criteria formulated in Sect. 2.2.4. Evidently, both the observed and simulated (with the BB emissions) data show rather similar spatial patterns, indicating the presence of heavy smoke plumes over many areas both in western Siberia (in particular, between Omsk and Krasnoyarsk) and eastern Siberia (south-east of Yakutsk). Importantly, the effects of the same fires can be readily seen in both the AAOD and AOD data. The differences between the satellite data and simulations are also considerable (the root mean square errors normalized to the mean values equal 0.49 and 0.46 in the cases of the AAOD and AOD distributions, respectively). In particular, both AAOD and AOD tend to be overestimated by the model in Central Siberia and the Far East but underestimated in Western Siberia. These differences may be due to a variety of reasons, including errors in the spatial allocation and the magnitude of fire emissions, uncertainties in the satellite retrievals, as well as the model's inability to take into account spatial and temporal variations in the optical properties of the actual BB aerosol.

Figure 6 presents the temporal (daily) variations in the spatially-averaged AAOD and AOD data. Taking into account that the number of the spatially resolved data points averaged over a given day strongly varies from day to day and that the agreement between the daily mean data from the simulations and observations is likely to degrade on days with a small amount of available data, we required that each observational data point (and its simulated matchup) shown in Fig. 6 was composed of at least ten values corresponding to different grid cells. Otherwise, an observational data point was considered as an outlier. These outliers were not included in Fig. 6 and disregarded in the comparison statistics (reported in the legend of Fig. 6); the corresponding simulated values (shown in Fig. 6) were averaged over the whole study region. The results presented in Fig. 6 indicate that when the model used the optimized emissions, it reproduced the daily variations both in AAOD and AOD reasonably well, with correlation coefficient values of about 0.8 and very small biases that do not exceed 5%. The agreement of the simulations is evidently better with the AOD observations than with the AAOD retrievals. This is an expected result, given the fact that both the OMI-derived AAOD data and the corresponding simulations are likely to have larger uncertainties than the observations and simulations of AOD. The correlation coefficient values were considerably smaller and biases were much larger when the model used the "initial-guess" BB emissions calculated with the correction factors equal unity. These findings indicate that the inversion of the AAOD and AOD observations results in major improvements in the model performance.

Figure 7 compares the relationships between AAOD and AOD according to the satellite observations and our simulations. The relationships include all of the gridded daily data points selected for the validation dataset. As follows from Eq. (4), the relationship between AAOD and AOD is indicative of the EC/OC ratio in BB aerosol particles. Therefore, the adequacy of

the relationship between the modeled AAOD and AOD values is an important pre-requisite for accurate estimations of the EC/OC ratio in the BB aerosol emissions. Figure 7 also shows the similar relationships between the AAOD and AOD data derived from the AERONET measurements, which were used to evaluate the parameters of Eq. (4), and between their modeled counterparts.

Evidently, although the model cannot explain some strong variations in the AAOD/AOD ratios derived from the observations (which may be enhanced due to temporal and spatial inconsistencies between the OMI and MODIS measurements), it reproduces the "observed" relationship quite well on average. Specifically, both the observations and simulations indicate that the ratios of the average values (indicated by angle brackets) of AAOD and AOD, as well as the slopes of the regression lines fitted to the AAOD and AOD values, are close to 0.1 (±11%) According to Eq. (4), this value corresponds to an EC/OC

ratio of about 0.045, which is rather similar to that of 0.052 assumed in the GFED4 inventory for BB emissions in extratropical forests. Similar values of the <AAOD>/<AOD> ratio are characteristic for the AERONET data and their simulated matchups, although the latter is slightly biased positively. The consistency between the <AAOD>/<AOD> ratios in the satellite observations and AERONET data can be considered as evidence that the measurements of the optical properties of BB aerosol at the AERONET sites are sufficiently representative of the typical optical properties of BB aerosol in the whole study

region. Note that the cluster of green points above the regression line in Fig. 7b indicates a distinct contribution of the agricultural and grass fires featuring much larger ratios of the BC and OC emission factors (see Table 1) than the predominant forest fires.

### 3.3 Evaluation of the simulated BC and OC concentrations against observations at ZOTTO

Figure 8 illustrates the evaluation of the simulated concentrations of EC and OC against the corresponding in-situ observa-

tions at the ZOTTO site, which in summer 2012 was surrounded by numerous fires (see Fig. 2). The observational data points shown in Fig. 8 represent EC or OC concentrations detected in the individual aerosol filter samples. The simulated data from the CHIMERE model, which was run both with the optimized and initial-guess BB emissions as well as without BB emissions, were averaged over each individual sampling period, which was of different length for different samples (see Sect 2.1.5). The comparison statistics, including the mean value, the difference between the mean values of the simulated

and observed data (the bias) along with the 90 % confidence interval, and the correlation coefficient are reported for each simulation in the legends of Fig. 8. Note that to the best of our knowledge, aerosol simulations performed with a chemistry transport model have never been evaluated previously against EC and OC measurements in Siberia.

Both the EC and OC concentrations predicted by the model for the "base-opt" and "base-ini" cases correlate very well (r > 0.9) with the corresponding observations. The EC concentrations are on average somewhat overestimated in the simulations

with the optimized emissions: the agreement of the mean concentrations would be perfect if the simulated concentrations were reduced by 23 %. However, a predominant part of this difference between the mean simulated and measured concentrations can be explained by random model errors. A remaining smaller part of the difference may be explained by the uncer-

tainty in our estimates of the emission correction factors (and thus in BC emissions specified in the model), which is about 35 % (see Sect 3.1). The EC concentrations in the simulation with the initial-guess emissions are, on the other hand, a factor of 1.32 too low on average. The fact that the "base-ini" simulation demonstrates a slightly better performance in terms of the correlation coefficient than the "base-opt" simulation may be indicative of a smaller monthly variability of the BC emission

and/or conversion factors representative of the forest fires, which predominate in the vicinity of the ZOTTO site, compared to the variability of the same parameters representative of the fires across the whole study region.

The OC concentrations simulated with the optimized emissions appear to be slightly biased low, but the available bias estimate is not statistically significant. In contrast, the OC concentrations are strongly (by more than a factor of 2) underestimated in the "base-ini" simulation: this result is consistent with a similar underestimation of AOD in the same simulation

(see Fig. 6) and further supports our finding (see Table 2) that the initial-guess OC emissions should be strongly increased. Note that according to our "bgr" simulation, i.e., if BB emissions in Siberia were completely absent, both EC and OC concentrations at the ZOTTO site would be more than an order of magnitude lower than observed,. This fact indicates that possible uncertainties in anthropogenic EC emissions are not likely to be responsible for any noticeable bias in the EC concentrations simulated with BB emissions. Overall, the above comparison indicates that our top-down BB EC and OC emission es-

timates, considered together with their confidence intervals, are consistent with the EC and OC observations made in Central Siberia.

### 3.4 Comparison of the simulated data with aircraft measurements

Figure 9 shows the tracks of the flights performed in the framework of the YAK-AEROSIB campaign in July-August 2012. The northern and southern sectors of the trajectory correspond to the flights performed on 31 July and 1 August, respective-

ly. The flight tracks are overlaid onto the grid of our model and are shown along with the observed and simulated values of the CO mixing ratio, which were averaged over the region covered by each grid cell that had been intersected by the aircraft trajectory. One can notice several grid cells (north and south of Krasnoyarsk and around Yakutsk) where the CO mixing ratios (both in the measurements and in the simulations) exceed 400 ppb. These "hot spots", corresponding to high percentiles of the CO mixing ratio, were not found in the respective data from the "bgr" simulation (which are not shown in Fig. 9) and

thus are likely due to BB emissions. Note that crossing BB smoke coinciding with high CO plumes has been confirmed by direct visual/olfactory evidence as well as by a clear increase in the K+ ion concentration in the forest fire plumes (Antokhin et al., 2018). Taking these considerations into account, we used high percentiles of the CO observations to pinpoint occurrences when the aircraft traversed BB plumes.

Specifically, we selected $PM_{2.5}$ and BC (eBC) measurements matching the CO mixing ratios exceeding the 90[th] percentile

(395 ppb) or 80[th] percentile (277 ppb) of the distribution of the CO mixing ratios. The average CO mixing ratios in the selected subsets of the measurement and simulated data were, respectively, 602 ppb and 374 ppb for the 90[th] percentile and 465 ppb and 311 ppb for the 80[th] percentile. As the selection criterion was applied only to the observational data that manifest strong subgrid variability, the fact that the average CO mixing ratios are larger in the observations than in the simulations

does not necessarily mean that the model underestimates the CO mixing ratios in the BB plumes. More importantly, the corresponding average CO mixing ratios simulated without BB emissions (110 and 111 ppb for the selection criteria based on 90[th] and 80[th] percentiles, respectively) are much smaller than those simulated with BB emissions: this fact confirms that the BB plumes observed during the YAK-AEROSIB campaign are reasonably well matched in the simulations by large concen-
trations of CO originating from vegetation fires.

Figure 10 shows the relationships between the $PM_{2.5}$ and BC mass concentrations selected as explained above. To evaluate these relationships, they were fitted with linear regressions without intercept; the fit equations are reported in the legends of Fig. 10. Assuming that the contribution of the background aerosol fraction to the selected BC and $PM_{2.5}$ measurements was negligible, we regard the value of the slope of the best fit line as an estimate of the BC-to-$PM_{2.5}$ ratio in BB aerosol measured
during the flights. Note that according to our simulations, the background BC and $PM_{2.5}$ concentrations corresponding to the selected measurements were, on average, very small (only 0.02 and 14 $\mu$g m$^{-3}$, respectively, for the selection criterion based on the 80[th] percentile) compared to the range of the values presented in Fig. 10. The slopes of the fits to the observational data are about 0.021 for any of the two selection criteria considered. This value is in the middle of the range of the eBC/$PM_{2.5}$ ratio values (0.01-0.045) observed in Siberian smoke plumes earlier (Kozlov et al., 2008). For comparison, the
BC/$PM_{2.5}$ ratio for fresh BB aerosol in extratropical forest is assumed to be 0.033 in the GFED4 inventory (van der Werf et al., 2017); that is, a factor of 1.5 larger than the value found in this study.

The large scatter of the experimental data points may reflect the actual variability of the BC/$PM_{2.5}$ ratios in BB aerosol particles sampled by the aircraft instruments, although it may also be due to the measurement uncertainty, including temporal mismatches between BC and $PM_{2.5}$ measurements. The emissions from the flaming and smoldering phases of fires have very
different BC/$PM_{2.5}$ ratios, and an aircraft flying through plumes near the fires often passes through sub-plumes originating from the different fire phases and thus having very different compositions. After some transport, the smoke from the flaming and smoldering parts of fires becomes well mixed in the plumes. This may explain why the scatter is smaller in the relationships between EC and OC concentrations in the BB aerosol samples collected at the ZOTTO site (see Mikhailov et al., 2017 and Fig. 9a therein). In contrast, the scatter of the simulated data points is very small. The variability of the BC/PM ratios
may be strongly underestimated in our simulations as a result of the simplistic model representation of the complex patterns of spatial and temporal variability of BB BC and $PM_{2.5}$ emissions and also due to the probably inadequate representation of the BB aerosol aging processes in CHIMERE.

The BC/$PM_{2.5}$ ratio in our simulations is about 3 % and 10 % larger than the corresponding estimate derived from the YAK-AEROSIB measurements with the selection criteria based on the 90[th] and 80[th] percentiles of the CO mixing ratio, respective-
ly. As the eBC concentrations measured with an aethalometer are likely to be different from EC concentrations measured with a thermo-optical method (see Sec. 2.1.6), these differences are not indicative of any biases in our estimates of BC emissions (which are evaluated in this study as emissions of EC). Furthermore, any discrepancy between the slopes of the best

fits to the observational and simulation data could easily be eliminated by decreasing the correction factors $F^{BC}$ (and thus BC emissions) in the simulations within the uncertainty range of the optimal estimates of $F^{BC}$.

Unfortunately, due to the absence of frequent measurements of an independent tracer of biomass burning in the YAK-AEROSIB observations, we could not use them for evaluation of model predictions of the absolute values of BC and PM$_{2.5}$ concentrations. Unlike the measurements at ZOTTO, which were performed in an almost pristine environment, the aircraft trajectory during the YAK-AEROSIB campaign passed over polluted areas near large cities, where the contributions of anthropogenic sources to the BC and PM$_{2.5}$ concentrations could be considerable or even predominant. Nonetheless, we could compare our simulations with the campaign-average concentrations. Accordingly, we found that the average BC and PM$_{2.5}$ concentrations were 0.62 and 22 $\mu g\ m^{-3}$ in the observations, while the average of their simulated matchups were 0.44 and 28 $\mu g\ m^{-3}$. In view of the potentially large measurement uncertainties as well as the limited representativeness of the aircraft measurements at the scales resolved in our simulations, the differences between these average concentrations cannot be considered as clear evidence for biases in either BB or anthropogenic emissions specified in our model. Overall, the comparison of our simulations with the YAK-AEROSIB data shows a reasonable agreement, although it also highlights the difficulties and uncertainties associated with validation of BC simulations against the optical measurements of aerosols.

## 3.5 BC and OC emission estimates

Figure 11 shows the spatial distribution of the average BB BC emissions calculated for the study period in accordance with Eq. (2) using the MODIS FRP data and optimal estimates of the correction factors (see Sect. 4.1) constrained with the OMI AAOD and MODIS AOD retrievals. Not surprisingly, the distribution of BC emissions generally replicates the spatial patterns of FRP (see Fig. 2) and is also similar to the distributions of AOD and AAOD shown in Fig. 5. Grid cells with strong BC emissions cover vast areas in Western and Central Siberia, as well as in Eastern Siberia (east of Yakutsk and along Russia's border with China). For comparison, Fig. 11 also shows the corresponding spatial distributions based on the data from the GFED4.1s and FEI-NE emission inventories. All the distributions look rather similar, although there are also many differences between them. Most of the differences appear to have a random character, but it is noticeable that the emissions obtained in this study and based on the FEI-NE data tend to be stronger in many "hot spots" than those based on the GFED data. Greater FEI-NE BC emissions compared to those from GFED4 can be explained by an almost a factor of two difference in the BC emission factors assumed in FEI-NE and GFED4, as well as by differences in the methodologies to estimate fuel loadings. Similar reasons (that is, biases in the emission factors and/or in the fuel consumption estimates involved in the GFED4 inventory) may be behind the differences between the GFED4 data and our estimates. It is also noticeable that a much larger number of grid cells in the distributions based on our estimates are associated with relatively weak emissions in the range from 0.01 to 0.05 $g\ m^{-2}$ than in the distributions based on both the GFED and FEI-NE data. This difference indicates that emissions from some small fires (especially in agricultural areas) may be missing in the GFED and FEI-NE inventories (based on the burnt area data) but are taken into account in our calculations based on the FRP measurements.

Figure 12 and Table 3 report our top-down estimates of the total monthly BC emissions from fires in the study region, as well as the estimate of the integral BB BC mass emitted in the study region in May-September. The uncertainties of our estimates are reported in terms of the 90[th] percentile confidence level. Our estimates are shown in comparison with the corresponding values calculated using the GFED4 and FEI-NE emission data. The uncertainty level in the GFED4 data has not

been reported, and therefore it is not indicated in Fig. 12. Note, however, that previous studies in which AOD simulations based on the GFED inventory were evaluated against corresponding observations in different regions of the world (see, e.g., Tosca et al., 2013; Reddington et al., 2016; Petrenko et al., 2017) indicated that the GFED data for BB aerosol emissions may be very uncertain, such that they need to be corrected with adjustment factors sometimes exceeding 10 on a regional scale. The uncertainty reported for the FEI-NE data is 63 % (Hao et al. 2016). It has not been specified whether this uncer-

tainty characterizes gridded data or total regional emission estimates; we assume here that the latter is true.

According to our estimates, the fires in the Siberian study region released 405 (±135) Gg of BC during the study period. This value is many times larger than the total BC amount (25 Gg) that was emitted from other sources in the study region and period, according to our calculations based on the data of the ECLIPSE V5 emission inventory for 2010 (Klimont et al., 2017). For comparison, our estimate of the total BB BC emissions is also much larger than the total annual anthropogenic

BC emissions in North America (249 Gg yr$^{-1}$) and less than a factor of two smaller than the total annual anthropogenic BC emissions in Europe and Russia (660 Gg yr$^{-1}$) in 2010 (Klimont et al., 2017). About 40 % (139 Gg) of the total amount of BC released from the fires during the whole study period was emitted in July. The emissions were smallest in September (20 Gg) and ranged from 71 to 96 Gg in May, June, and August. Note again that BC emissions are evaluated in this study as emissions of EC.

Our estimates indicate that the total BC emissions from Siberian fires in the period considered are strongly underestimated in the GFED4 inventory (by more than a factor of 2), in which these emissions are estimated at about 198 Gg. Taking into account that GFED is widely used as a "reference" database for estimations of atmospheric and climatic effects of open biomass burning, we believe that this is a significant finding. The relative difference between our monthly BC emission estimates and the corresponding GFED4 data is largest in September, exceeding a factor of 8; it is also large (a factor of 3) in

May. In contrast, the BC emissions in the FEI-NE inventory (614 Gg) are larger than ours, although this difference is not significant in view of the reported uncertainty in the FEI-NE data. Note that, while no specific measure (EC, EBC, rBC) is identified for BC in the GFED4 inventory, Hao et al. (2016) specified that the FEI-NE inventory employed emission factors for refractory BC (rBC). However, we are not aware of any procedure that could allow us to adjust for the differences between rBC and EC. Overall, our top-down estimates provide a compromise between the data of the GFED4 and FEI-NE in-

ventories. Importantly, the evaluated uncertainty in our estimates is much smaller than both the differences between the estimates based on the two inventories considered and the reported uncertainty of the FEI-NE data. Therefore, the satellite data provide stronger constraints on BC emissions from Siberian fires, compared to the state-of-the-art emission inventories.

Although the estimation of OC emissions was not the focus of our study, our top-down estimates of the OC emissions (see Fig. 13a and Table 3) are useful to consider here, as they allow us to further evaluate the overall integrity of our method and results. We also report BC/OC emission ratios (see Fig. 13b) calculated as the ratio of our estimates for BC and OC emissions along with the emission ratios calculated using the GFED4.1s data. Note that FEI-NE does not provide data on OC emissions.

The results shown in Fig. 13a indicate that the pattern of monthly variations of OC emissions is not very similar to that of BC emissions. Specifically, the OC emissions in May are found to be much smaller than in June, while the BC emissions were larger in May (see Fig. 12). But similar to our BC emission estimates, our estimates of OC emissions are much larger than the corresponding estimates based on the GFED4 data. Our estimate for the integral emissions over the fire season considered is a factor of 2.2 larger than the corresponding estimate based on the GFED4 data. Based on comparison of satellite-derived and simulated AOD, several previous studies showed evidence that OC emissions provided by the GFED inventory may indeed be underestimated in different regions of world, including Siberia (see, e.g., Petrenko et al., 2012; 2017; Tosca et al., 2013; Konovalov et al., 2014; 2015; Reddington et al., 2016), although it was also argued (Konovalov et al., 2015; 2017b) that models may underestimate AOD due to inadequate representations of the BB aerosol aging processes. So it is possible that a part of the differences between our optimal estimate of the OC emissions and the corresponding GFED data may compensate for some missing processes (e.g., involving the formation of SOA due to oxidation and condensation of semi-volatile organic compounds) in our model.

In spite of the very significant differences of our BC and OC emission estimates with respect to the GFED4 data, the BC/OC emission ratio ($0.046\pm0.014$ g g$^{-1}$) obtained in our analysis (see Fig. 13b) is consistent, in the case of the integral emissions for the study period, with that in GFED4 ($0.054$ g g$^{-1}$). Furthermore, the monthly variations of the BC/OC emission ratio according to our estimates are qualitatively similar to those according to the GFED4 data. Specifically, both the GFED4 inventory and our estimates indicate that the BC/OC emission ratio was bigger in May and September than in the summer months. However, our estimates also indicate that the BC/OC emission ratio may be underestimated by GFED4 in May and overestimated in the summer months. Monthly variations of the BC/OC emission ratio in the GFED inventory are a result of changes in the presumed fire fuel: in particular, the monthly variations shown in Fig. 13b indicate that, according to the GFED4 data, the contributions of agricultural and grass fires to the BB BC emissions were slightly bigger in May and September than in the summer months. The same factor can explain (at least partly) the monthly variations in our estimates of the BC/OC emission ratio. To illustrate this point, Figure 14 shows the spatial distributions of the relative contribution of agricultural/grass fires to BB BC emissions integrated over a month, along with the spatial distributions of the corresponding BB BC emission values. The distributions were obtained for three different months (May, July, and September) using Eq. (2) and the optimal estimates of the correction factors, $F^{BC}$. Evidently, the fires that occurred in the study region in May burned mostly in agricultural lands and grasslands, even though the BC emissions from intensive forest fires were also quite significant in several grid cells. In contrast, forest fires were clearly predominant in July (as well as in the other summer months).

Unlike the situations in both May and July, BB BC emissions in September were not clearly associated with any predominant fuel category: along with agricultural/grass fires in the southwestern and southern parts of the study region, there were relatively strong forest fires north of Tomsk and Krasnoyarsk and west of Yakutsk. Taking these observations into account, it can be speculated that the big difference between the BC/OC emission ratios in July and September is, to some extent, a manifestation of the diversity of fire regimes across the boreal region (Conny and Slater, 2002). Note that the spatial distribution of our emission data is insufficient to enable distinguishing between agricultural and grassland fires. However, according to the GFED4 inventory, agricultural burns strongly dominate over grass fires both in May and September (by a factor of 5 at least).

It is noteworthy that our estimates of the BC/OC emission ratios (in the range from 0.036 to 0.042 g g$^{-1}$) for the summer months are only insignificantly – taking into account the confidence intervals – different from the EC/OC ratio of 0.038 g g$^{-1}$ that was derived for BB aerosol by Mikhailov et al. (2017) from aerosol measurements at ZOTTO in summer. Furthermore, our estimate for May (0.093±0.03 g g$^{-1}$) is in a good agreement with the EC/OC ratio (0.08±0.02 g g$^{-1}$) found by Mikhailov et al. (2017) for BB aerosol originating predominantly from agricultural fires in spring. As SOA formation simulated with the "standard" aerosol module of CHIMERE contributes very insignificantly to BB aerosol concentrations (Konovalov et al., 2015; 2017b), the ratios of the BC and OC emissions specified in our simulations are quantitatively almost the same as the simulated BC/OC ratios in the ambient aerosol particles, irrespective of their age. Therefore, our estimates of the BC/OC emission ratios look reasonable in view of the independent ambient observations in central Siberia. This finding confirms the validity of our estimation method and the obtained results.

### 3.6 Sensitivity tests

The confidence intervals for our optimal estimates of BC emissions (Table 3 and Fig. 12) do not necessarily include possible uncertainties and biases that may be associated with systematic model errors and data selection criteria. Based on our understanding of likely reasons for such uncertainties and biases, we specified eight sensitivity tests listed in Table 4. The sensitivity analysis was focused on the estimation of the total BC emissions over the study period. The correction factors $F^{BC}$ and $F^{OC}$ for each test case were obtained by applying Eq. (9) and (10) to the optimal ("base case") estimates of the correction factors (see Table 2). One more iteration of the estimation procedure was sufficient to obtain the test estimates of the total BC emissions with a relative numerical error of 3 % or less. The total BC emission estimates for each case and the relative differences with respect to the base case estimate reported in Table 3 are also listed in Table 4.

Test case No. 1 addresses systematic differences between the photochemical ages of BB aerosol observed at the AERONET sites that provided the data considered in our analysis (see Sect. 2.4) and those of BB aerosol observed by satellites. Figure 15 shows the histograms of the photochemical ages estimated in accordance with Eq. (1) separately for the AERONET data and for the satellite data from the datasets selected for our analysis. Compared to BB aerosol observed from satellites (which have a median photochemical age of 15.4 h), the BB aerosol at the AERONET sites was typically more aged (with a median photochemical age of 25.8 h). If the relationship given by Eq. (4) is sensitive to the photochemical age of the aerosol, these

differences can result in some bias in the modeled AAOD values. To get an idea about the significance of such bias, we disregarded satellite data corresponding to photochemical ages smaller than 11 h. The remaining satellite data have approximately the same median photochemical age as the AERONET data. This restriction resulted in a small change of the optimal BC emission estimate, which increased by less than 3 %. This result does not necessarily mean that the BB aerosol composition and its optical properties are not strongly affected by aging; rather it may mean that changes of AOD and AAOD, as well as those of the monthly BC emission estimates due to aerosol aging, tend to compensate each other in the total BC emission estimate.

The impact of aerosol aging on our estimates is further addressed in test case No. 2. Specifically, the goal of this test case is to assess a potential bias in our BC emission estimates due to a probable underestimation of the SOA contribution to aged BB aerosol. To this end, we performed a simulation in which the yields of all SOA species from oxidation of major volatile SOA precursors (such as toluene, xylenes, isoprene and terpenes) were enhanced by a factor of 7 with respect to the "base-opt" simulation, while the reaction list and the reaction rates (as well as all other simulation settings) were kept unchanged. As a result of this model modification, a relative enhancement of the averaged (over the whole period and region considered) POM column amounts due to SOA formation increased from only 2.6 % (in the "base-opt" case) up to 27 % (in the test case). Note that the SOA enhancement increased more strongly than the SOA yields, probably because some of the SOA species are assumed to be semi-volatile in CHIMERE, and thus their condensed fraction increases with their total concentration. The increased SOA contribution to POM in our test case simulation corresponds to the upper margin of the wide range of the BB POM enhancements observed in aging BB plumes in several field studies in North America (Cubison et al., 2011). In spite of the considerable changes in the simulated POM fields, the optimal BC emission estimate changed only insignificantly, increasing only by 4.6 % (see Table 4). This result is in line with the above discussion of the robustness of our BC emission estimates (see Sect. 2.2.4) with respect to the treatment of POM aging in our model. It is not surprising, however, that our estimates of the correction factors $F^{OC}$ and of the OC emissions were found to be more sensitive to the changes in the POM simulations; specifically, the top-down estimate of the total OC emissions dropped by ~15 % in the test case as a result of the increases in aerosol abundances and of a slight decrease in the mass extinction efficiency.

Test case No. 3 addresses the uncertainty associated with the representation of wet deposition of BB aerosol particles in our simulations. As noted in Sect. 2.2.1, we assumed that BB aerosol particles are hydrophobic and therefore are not susceptible to in-cloud scavenging; accordingly, the empirical uptake coefficient in the base case simulations with BB emissions was set to be zero. This assumption can result in overestimation of the life-time of BB aerosol particles, which tend to become more hydrophilic as the aerosol ages (Paramonov et al., 2013), and, consequently, in a negative bias of our BC emission estimates. To get an idea about the magnitude of this possible bias, we performed a test simulation (with the optimized BB emissions) in which the empirical uptake coefficient was set to be unity; this setting corresponds to the assumption that BB aerosol particles are hydrophilic. The use of this simulation instead of the original simulation with the optimized BB emissions in our estimation procedure resulted only in a minor increase (~6 %) in our optimal estimate of the total BB BC emissions; the

changes in the monthly estimates are found to be similarly small. This is an expected result, as the major fires considered in our analysis occurred mostly during dry periods with low precipitation. Therefore, the test case No. 3 indicates that probable changes in hygroscopicity of ambient aerosol particles due to BB aerosol aging processes will not significantly affect our BC emission estimates.

Test cases No. 4 and No. 5 are designed to evaluate to which extent our top-down BC emission estimate can be affected by a possible bias in the background AOD values predicted by CHIMERE. To get an idea about such a bias, we followed the approach suggested by Konovalov et al. (2014). Specifically, we first selected the days and grid cells (irrespective of the availability of AAOD data) in which the MODIS-retrieved AOD data are available and the contribution of fires to the modeled AOD values (corresponding to the selected the days and grid cells) does not exceed 10 % of the background AOD values,

and then we evaluated the mean difference between the MODIS-retrieved and modeled AOD for these selected data points. We found that the mean value for the modeled AOD (~0.17) is considerably higher than the mean value (~0.10) for the observed AOD. Taking into account that the bias in the background AOD values in pixels affected by fires may be somewhat different from that representative of background conditions, we considered larger changes in the background AOD by increasing or decreasing it by 50 %. The test results indicate that a probable positive bias in the background AOD values is

associated with some underestimation (by less than 20 %) of BC emissions in our procedure; if the bias were absent, the difference between the BC emission estimates inferred from the satellite observations and those calculated with the GFED4 data would be even larger than in the base case. The sensitivity of the optimal estimate is strongly asymmetric with respect to the enhancement and reduction of the background AOD: this is probably due to an impact of the changes in the background AOD on the selection of data according to the criterion given by Eq. (8).

Test case No. 6 addresses the uncertainties associated with the background AAOD. On the one hand, the AAOD data have been retrieved from the OMI measurements under the assumption that each observed pixel is characterized by only one type of aerosol. Consequently, the absorption caused by other types of aerosol has effectively been disregarded, although it might actually affect the AAOD retrievals. Thus, to prevent overestimation of the BC emissions, the background AAOD (predicted by CHIMERE) was subtracted from the AAOD retrievals as suggested by Eq. (7). On the other hand, BB plumes are typical-

ly reaching much higher altitudes than anthropogenic aerosol: this is taken into account in the OMAERUV retrieval algorithm by assuming that the vertical distribution of urban/industrial aerosol is largest at the surface, while the concentration of carbonaceous aerosol in smoke layers at mid- and high-latitudes typically peaks at 6 km. The AAOD values retrieved by assuming that the aerosol layer is residing near the ground are much larger than those corresponding to the assumed heights of 6 km or even 3 km. So, if the aerosol in a given pixel is identified as carbonaceous BB aerosol, a part of AAOD corres-

ponding to the anthropogenic aerosol is likely to be underestimated in the retrievals. Therefore, a simple subtraction of the background AAOD values from the AAOD retrievals may result in an underestimation of the BC emissions in our analysis. To get an idea about the maximum magnitude of this underestimation, the background AAOD values were entirely disregarded in test case No. 6. The test result indicates that the underestimation is probably rather small (less than 10 %); note,

however, that it may actually be larger if the background AAOD values in our simulations are biased high. Unfortunately, we cannot properly evaluate the possible overestimation of the BC emissions in the case where the background AAOD is strongly underestimated. However, as noted above, the simulated background AOD is overestimated, so it seems reasonable to assume that the background AAOD is overestimated, too. Accordingly, we believe that the uncertainty of the best estimate of the BB BC emissions with respect to the intrinsic uncertainty associated with the background part of the AAOD retrievals is likely within the difference between the estimates given by the "base-opt" case and test case No. 6.

As noted above (see Sect. 2.2.1), the AAOD retrievals corresponding to different assumed altitudes of the aerosol center of mass were selected in our analysis by using the smoke layer heights derived from our simulations. Ideally, this approach ensures that the AAOD retrievals are consistent with the observed variations in the location and intensity of the fires. Nonetheless, in view of the possible uncertainties in the simulated vertical distributions of the BB aerosol, it is also useful to consider the BC emission estimates derived from the standard ("OMI_final") data product based on rather rough (climatological) estimates of the smoke layer heights. This is done in test case No. 7. We found that the standard data product yields a 22 percent lower BC emission estimate than the base case estimate. The difference between the two estimates is considerable, but it is still well within the uncertainty limits of the base case estimate.

Test cases No. 8 and No. 9 examine the sensitivity of our estimates to the selection criterion defined by Eq. 8. Specifically, we used a 50 percent higher and 50 percent lower values of $\gamma$ for test cases No. 8 and No. 9, respectively. The larger value of $\gamma$ selects data points with larger values of AOD and vice versa. The emission estimates obtained with a smaller value $\gamma$ are more prone to uncertainties associated with the background AOD. On the other hand, a stricter selection criterion results in a loss of information about relatively small fires. Anyway, the test results show that the sensitivity of our estimates to the big changes in $\gamma$ is relatively weak, suggesting that our base case estimates are sufficiently robust with respect to the selection criterion considered.

Finally, test case No. 10 is designed to address a potential issue concerning the representativeness of the OMI retrievals in view of the rather coarse resolution of our simulations. It seems reasonable to expect that when, for example, only one AAOD observation corresponding to BB aerosol is available for a given grid cell, the mean observed AAOD value inferred in our procedure for this grid cell is likely to be overestimated, as AAOD over the rest of the grid cell's area may be much smaller. However, the overestimation can hardly be very large for the very intense and widespread Siberian fires considered, because, in this case, the smoke plumes are likely to cover a large fraction of the grid cell area. To examine this issue, we disregarded any gridded AAOD data points that comprised less than 10 different AAOD observations (data pixels), while the maximum number of the pixels per grid cell in the data considered equals 26. Contrary to our expectations, we found that the estimate obtained in test case No. 8 is larger (by 17 %). This increase is found to be mostly due to an increase in the optimal estimates of $F^{OC}$. Apparently, the above limitation resulted in selection of MODIS AOD data that are more representative of grid cells affected by major fires and are matched by smaller AOD values simulated for the base case. Therefore, the result of this test is not indicative of any representativeness issue for the available OMI retrievals.

In general, the results presented in this section demonstrate that our estimate of the total BC emissions from Siberian fires is sufficiently robust with respect to possible uncertainties in the input data and the choices made in the estimation procedure. In particular, these results strongly support our findings that the GFED4 inventory significantly underestimates the BC emissions from Siberian fires.

## 3.7 BC transport into the Arctic

As argued in the introduction, studying BB BC emissions in Siberia is stimulated by the need to properly evaluate the role of BC in Arctic climate change. Therefore, it is important to know not only the amount of BC emitted from the fires but even more so the amount of BC transported into the Arctic. Using the three-dimensional hourly fields of BC mass concentrations and of the meridional component of wind speed from our optimized simulations, we calculated the hourly BC fluxes from the study region across the polar circle (66°33′ N) and then integrated them over altitude (from the surface up to the model domain top coinciding, approximately, with the tropopause), longitude, and time on a monthly basis. The fluxes were calculated separately for BC emitted from fires and from anthropogenic sources. In this way, we evaluated the total masses of BB and anthropogenic BC transported from the study region into the Arctic each month (see Fig. 16). Note that a part of the BC mass transported into the Arctic may be transported out of it back to the study region (when the corresponding transport times are shorter than the typical lifetime of BB aerosol with respect to deposition); however, such backward transport of BC is mostly not being taken into account in our calculations, as the model domain does not extend to the whole Arctic. We also evaluated the BC transport efficiency, defined here as the ratio of the BC amounts transported to the Arctic to the corresponding amounts of BC emitted from Siberian fires. This definition is similar but not identical to that introduced in Evangeliou et al. (2016), where the transport efficiency was defined as the ratio between the mass of BC deposited in the Arctic and the mass of BC emitted from a given region.

Our estimates indicate that vegetation fires contributed a predominant part (as large as 95 %) of the integral BC mass transported into the Arctic from the study region during the five months considered (see Fig. 16a). This amount corresponds to an overall transport efficiency of about 27 % (see Fig. 16b): that is, about a quarter of the total BC emitted from Siberian fires was transported into the Arctic. This estimate of the transport efficiency is comparable with that (about 30 %) obtained by Evangeliou et al. (2016) for BC emitted from fires in Asia in the summer periods 2012-2013. Our results show that the transport efficiency was not constant across the different months. In particular, it exceeded 60 % in September and was less than 15 % in June. Interestingly, the total BB BC mass transported into the Arctic in September is found to be slightly larger than that in June (see Fig. 16a), in spite of the fact that the amount of BB BC emissions is more than a factor of 3 larger in June than in September. This fact emphasizes the potential climatic importance of the fires that occur in Siberia in early fall. According to our results (see Sect. 3.5), the BB emissions from these fires (which were most intensive within about 300-500 km west of Yakutsk, see Fig. 14f) are very strongly (by a factor of 8) underestimated in the GFED4 inventory (but note also that our BB BC emission estimate for September is very uncertain). By using the model run the for test case No. 3 (see Sect.

3.6) instead of the "base-opt" run, we made sure that disregarding the impact of BB aerosol aging on hygroscopicity of aerosol particles in our simulations did not have a significant effect on our analysis of BC fluxes. In particular, the overall transport efficiency evaluated under the assumption that BB aerosol particles are composed of hydrophilic material turned out to be only slightly smaller (25.2 %) than the corresponding base case estimate (27.6 %); among the individual months, the transport efficiency decreased most in May (from 29 % in the base case to 24 % in the test case). As a caveat, it should be noted that because of inter-annual meteorological variability, our monthly estimates of the transport efficiency in 2012 may not be applicable to other years. To improve the current understanding of the role of Siberian fires in the Arctic warming, the analysis suggested in this paper should be extended to a multi-annual period.

## 4 Summary and Conclusions

We have investigated the feasibility of constraining BC emissions from open biomass burning with AAOD retrievals from OMI satellite measurements by considering the case of the severe fires that occurred in Siberia in 2012. We developed an inverse modeling procedure enabling optimization of BB emissions based on MODIS FRP measurements by combining OMI AAOD retrievals and MODIS AOD data with simulations performed with the CHIMERE CTM. To limit possible errors in the simulated AAOD data due to uncertainties in the absorption properties of the BB aerosol, we employed an empirical parameterization predicting AAOD as a function of AOD and the ratio of BC and OC column densities. The parameterization is based on the experimental findings reported earlier (Pokhrel et al., 2016) and is fitted to data from two AERONET sites in Siberia; it assumes that the SSA of BB aerosol particles is a linear function of the elemental to total carbon ratio. As a result of the application of our inverse modeling procedure to the measurement and simulation data characterizing the BB aerosol in Siberia during the period from 1 May to 30 September, we evaluated the monthly correction factors for BB BC and OC emissions calculated using the FRP data and obtained top-down estimates of the total BC and OC amounts emitted each month in the period considered. Note that our estimation method implies that the BC emissions are evaluated as emissions of elemental carbon (EC) measured using a thermo-optical technique.

To validate the optimized BC and OC emissions, we used them to perform simulations that were evaluated against independent observational data. Specifically, we first compared our simulations with the OMI AAOD and MODIS AOD data that had been withheld from the optimization procedure. A reasonable agreement between the observations and simulations is found in the spatial distributions and daily time series of the both AAOD and AOD data. In particular, the correlation coefficients for the time series of spatially averaged AAOD and AOD values were found to be 0.79 and 0.84, respectively. Our simulations were further compared with in situ measurements of EC and OC mass concentrations at the top of the 300 m tower at the ZOTTO site (Mikhailov et al., 2017), situated at a remote location in central Siberia. Although the simulated EC concentrations turned out to be about 23 % larger than the observed ones, the bias was not found to be significant considering the uncertainties of our emission estimates and random model errors. A minor negative bias of about 7 % is found in the simulations of OC concentrations. It should be noted that unlike the satellite data, which cover the whole study region, the in

situ measurements of BB aerosol may contain some local features of fire regimes and fuels, which could not be reproduced in our simulations. We also compared our simulation with optical measurements of BC and $PM_{2.5}$ mass concentrations onboard an aircraft in the framework of the YAK-AEROSIB experiments. Due to a problem of distinguishing between the significant (on average) contributions of anthropogenic and BB sources to the measured aerosol concentrations, a direct com-

parison of the simulated and measured BC concentrations would not be sufficiently informative of the accuracy of our simulations of BB aerosol. Instead, we focused on a comparison of the relationships between the BC and $PM_{2.5}$ concentrations in the simulations and observations by using measurements of CO concentration to select the observations most representative of BB aerosol. The slopes of linear fits to the BC and $PM_{2.5}$ data from the simulations and observations are found to be in good agreement (within 10 %). This finding further confirms that the BB aerosol composition was simulated adequately.

We found that Siberian fires emitted $405 \pm 135$ Gg of BC in May -September 2012 (at the 90% confidence level). The BB BC emissions were largest in July, when $139 \pm 49$ Tg BC was emitted and smallest in September ($20 \pm 9$ Gg). Our estimates were compared to the corresponding estimates obtained from the GFED4 and FEI-NE databases. Our estimate of the total BB BC emissions in the study region and period is found to be a factor of 2 larger than the GFED4 estimate, but a factor of 1.5 smaller than the FEI-NE estimate. The differences of our monthly and season-total BC emission estimates with respect to

both GFED4 and FEI-NE data are statistically significant, although the differences with respect to the FEI-NE estimates are smaller than the large uncertainty range reported for the FEI-NE data.

The results of several sensitivity tests indicate that, although our estimates can be influenced to some extent by a number of factors associated, in particular, with data selection criteria and uncertainties in the simulations of optical properties of aerosol in the absence of fires, the possible bias in our estimate of the total BC emission is unlikely to exceed the estimated un-

certainty of about 35 %.

In spite of the significant differences between our BC emission estimates and the GFED data, the ratio of the total BC and OC emission estimates derived from the satellite data ($0.046 \pm 0.014$ g g$^{-1}$) is found to be consistent with the ratio of the corresponding BC and OC emission totals according to the GFED data ($0.054$ g g$^{-1}$). However, there are considerable differences between the BC/OC emission ratios obtained in this study and those calculated with the GFED4 data for the different

months. In particular, a larger value of the ratio of BC and OC emissions in May is found in this study ($0.093 \pm 0.030$ g g$^{-1}$) compared to that suggested by GFED4 ($0.06$ g g$^{-1}$): this difference may be indicative of an underestimation of BC emissions from agricultural burns and grass fires in the GFED4 inventory.

Finally, we estimated that about a quarter of the huge BC amount emitted from Siberian fires in May-September 2012 was transported across the polar circle into the Arctic. Therefore, the results of this study have a direct implication for reducing

major uncertainties associated with the current estimates of sources of BC in the atmosphere and snow/ice cover in the Arctic and for improving the general understanding of the role of BC in the Arctic climate system.

Overall, our analysis demonstrated that the OMI AAOD retrievals combined with the MODIS AOD data can provide useful constraints to the BB BC emissions. It is especially noteworthy that in the case considered in this study, the entire uncertain-

ty range for the BC emission estimates constrained with the satellite measurements turned out to be a factor of 1.5 smaller than the difference between the corresponding estimates provided by the two state-of- the-art emission inventories, GFED4 and FEI-NE. A major factor limiting the accuracy of the top-down estimates of BC emissions from Siberian fires is the uncertainty of the AAOD simulations. To reduce this uncertainty, more data of remote sensing and in situ measurements of

aerosol optical properties and composition (such as measurements of SSA and of the BC/OC and OC/POM ratios) in northern Eurasia are needed. Another significant uncertainty source in our estimates is associated with estimation of the altitude of the aerosol layer center of mass. Accordingly, future developments of our approach should include evaluation and optimization of the simulated vertical distribution of BB aerosol by using suitable satellite observations, such as, e.g., Cloud-Aerosol Lidar and Infrared Pathfinder Satellite Observations (CALIPSO) (Vaughan et al., 2004).

*Acknowledgements. The analysis of the satellite data performed in this study was supported by the Russian Foundation for Basic Research (grant no. 18-05-00911). The validation of the BC and OC emission estimates against the observations at ZOTTO was performed with a support from the Russian Science Foundation (grant agreement No. 18-17-00076). The data described in Sect. 2.1.5 were obtained within the SPBU BRICS grant (grant No. 11.37.220.2016). I.B. Konovalov acknowledges travel expenses in the framework of the PARCS (Pollution in the ARCtic System - PARCS) national project. M. O. An-*

*dreae and the research at ZOTTO are funded by the Max Planck Society. The authors acknowledge the free use of the AERONET data available from https://aeronet.gsfc.nasa.gov.*

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

**Table 1.** Emission factors (g kg$^{-1}$) for the carbonaceous components of BB aerosol. The numbers are adopted from the GFED4 inventory (van der Werf et al., 2017) and are based on Akagi et al. (2011) and Andreae and Merlet (2001) with subsequent updates.

|     | agricultural burning | grassland | forest |
| --- | --- | --- | --- |
| OC | 2.3 | 2.62 | 9.6 |
| BC | 0.75 | 0.37 | 0.5 |

**Table 2.** Optimal monthly estimates of the correction factors, $F^{BC}$ and $F^{OC}$, for the OC and BC emission rates (see Eq. 2), along with the ratios of optimal estimates for $F^{BC}$ and $F^{OC}$. The values in brackets indicate the 90% confidence intervals.

| Correction factor | May | June | July | August | September |
| --- | --- | --- | --- | --- | --- |
| $F^{BC}$ | 2.30(± 0.76) | 1.79 (±0.61) | 1.52 (±0.53) | 2.24 (±0.79) | 3.70 (±1.70) |
| $F^{OC}$ | 1.51 (± 0.48) | 2.27 (±0.55) | 2.28 (±0.57) | 2.73 (±0.78) | 2.14 (±1.02) |
| $F^{BC}/F^{OC}$ | 1.52 (± 0.49) | 0.79 (± 0.26) | 0.67 (± 0.21) | 0.82 (± 0.27) | 1.72 (± 0.64) |

**Table 3.** Optimal estimates of the BC and OC mass (Gg) emitted from fires in the study region for individual months of 2012 and for the whole period considered (1 May - 30 September). The numbers given in brackets are confidence intervals reported in terms of the 90[th] percentile.

| Species | May | June | July | August | September | All months |
|---------|-----|------|------|--------|-----------|------------|
| BC | 96.2 (± 32.0) | 71.4 (±24.3) | 139.0 (±48.4) | 78.9 (±27.9) | 19.8 (±9.1) | 405.3 (±134.6) |
| OC | 1032 (± 331) | 1774 (±430) | 3895 (±983) | 1889 (±538) | 254 (±121) | 8844 (±2197) |

**Table 4.** Estimates of the total BC emissions (Tg) from fires in the study region over the period from 1 May to 30 September 2012 for several test cases of the estimation procedure.

| test case No. | Brief description | Total BC emissions (Gg) | A relative difference with respect to the base case estimate, % |
|---------------|-------------------|-------------------------|------------------------------------------------------------------|
| 1 | BB aerosol photochemical age is larger than 11 h | 416 | +2.6 |
| 2 | the SOA yield in the simulations is increased by a factor of 7 | 424 | +4.6 |
| 3 | BB aerosol particles are assumed to be hydrophilic and affected by in-cloud scavenging | 431 | +6.3 |
| 4 | the background AOD is reduced by 50 % | 477 | +17.8 |
| 5 | the background AOD is enhanced by 50 % | 398 | −1.6 |
| 6 | the background AAOD is disregarded | 443 | +9.2 |
| 7 | the OMI "final AAOD" data product is used instead of the retrieval data provided for different aerosol layer heights | 317 | −21.8 |
| 8 | a weaker selection criterion ($\gamma$=0.5, see Eq. 8) is used | 477 | +17.8 |
| 9 | a stricter selection criterion ($\gamma$=1.5, see Eq. 8) is used | 388 | −4.1 |
| 10 | any gridded data point considered includes at least 10 AAOD pixels | 457 | +13.0 |

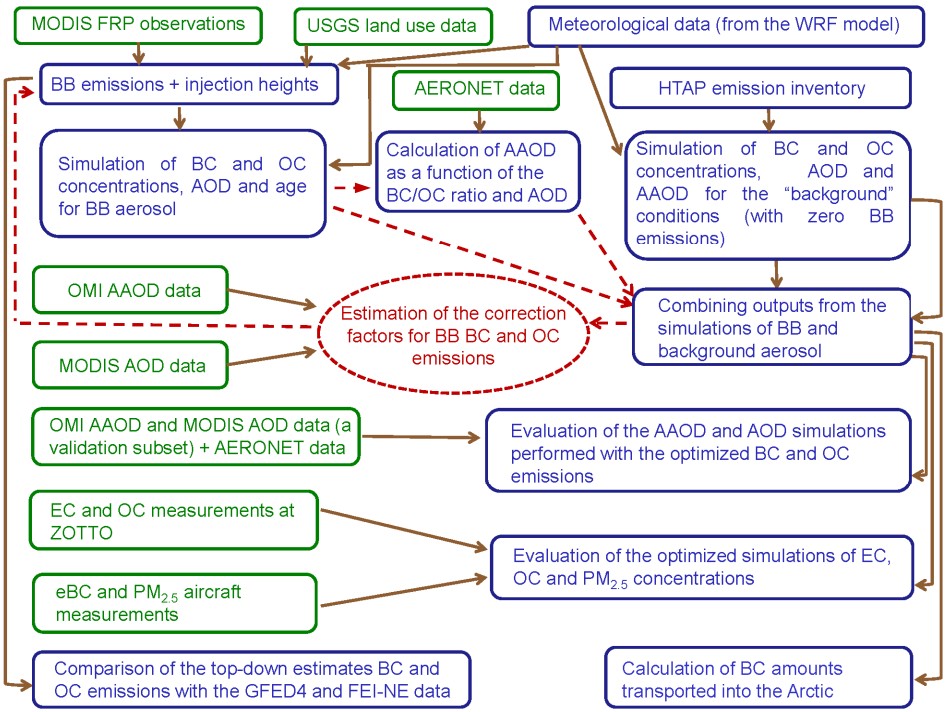

**Figure 1: Schematic representation of the study design. Green color is used to depict the observational data used in the analysis. Red color and dotted lines illustrate the iterative procedure aimed at optimization of the BB BC and OC emissions.**

**(a)**

**(b)**

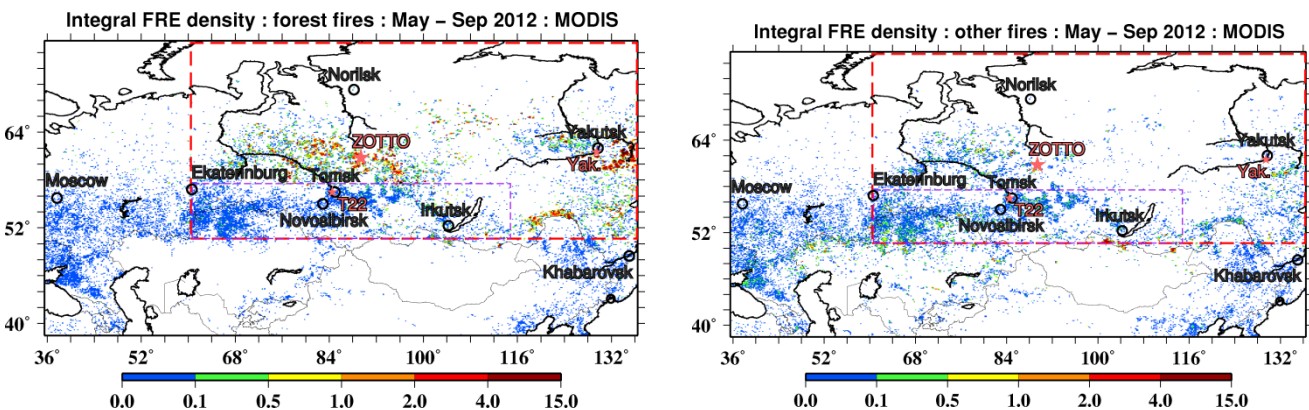

**Figure 2: Spatial distributions of the mean fire radiative energy density (MJ m$^{-2}$) in May-September 2012 for (a) forest fires and (b) other vegetation fires. The distributions with the spatial resolution of 0.2°×0.1° have been derived from the MODIS FRP data and are shown for the territories covered by a Siberian domain in the CHIMERE model. Red dashed rectangles depict the study region and short purple dashes indicate a supplementary sub-region discussed in Sect. 3.1. Pink asterisks indicate are the locations of the ZOTTO site and two AERONET sites, Tomsk-22 (T22), and Yakutsk (Yak.)**

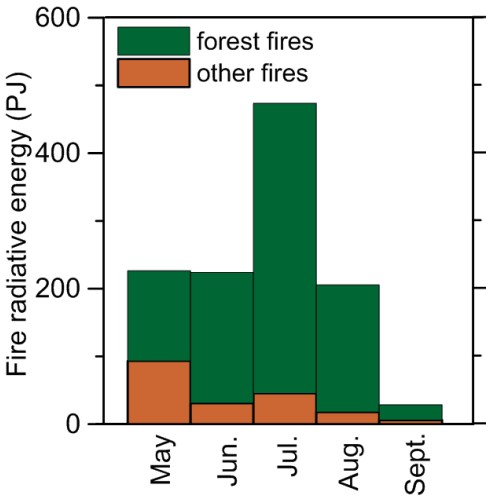

**Figure 3: Estimates of the total fire radiative energy (PJ) released from forest and other vegetation fires in the study region (see Fig. 2) in May-September 2012. The estimates have been obtained in this study from the MODIS FRP measurements as explained in Sect. 3.2.**

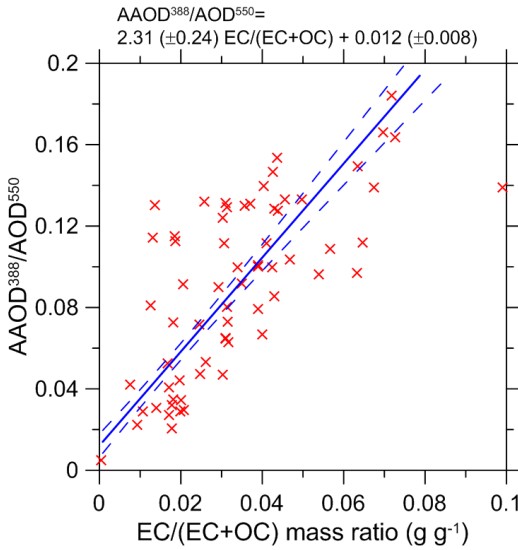

**Figure 4: The ratio of AAOD at 388 nm and AOD at 550 nm as a function of the ratio of elemental to total carbon in BB aerosol. Both ratios (depicted by red crosses) have been derived from observations at the Tomsk-22 and Ya-kutsk AERONET sites. A linear regression fitted with the ODR method and 1-sigma (68.3 %) confidence intervals of the fit are shown by solid and dashed blue lines, respectively. The best fit equation is given at the top of the figure.**

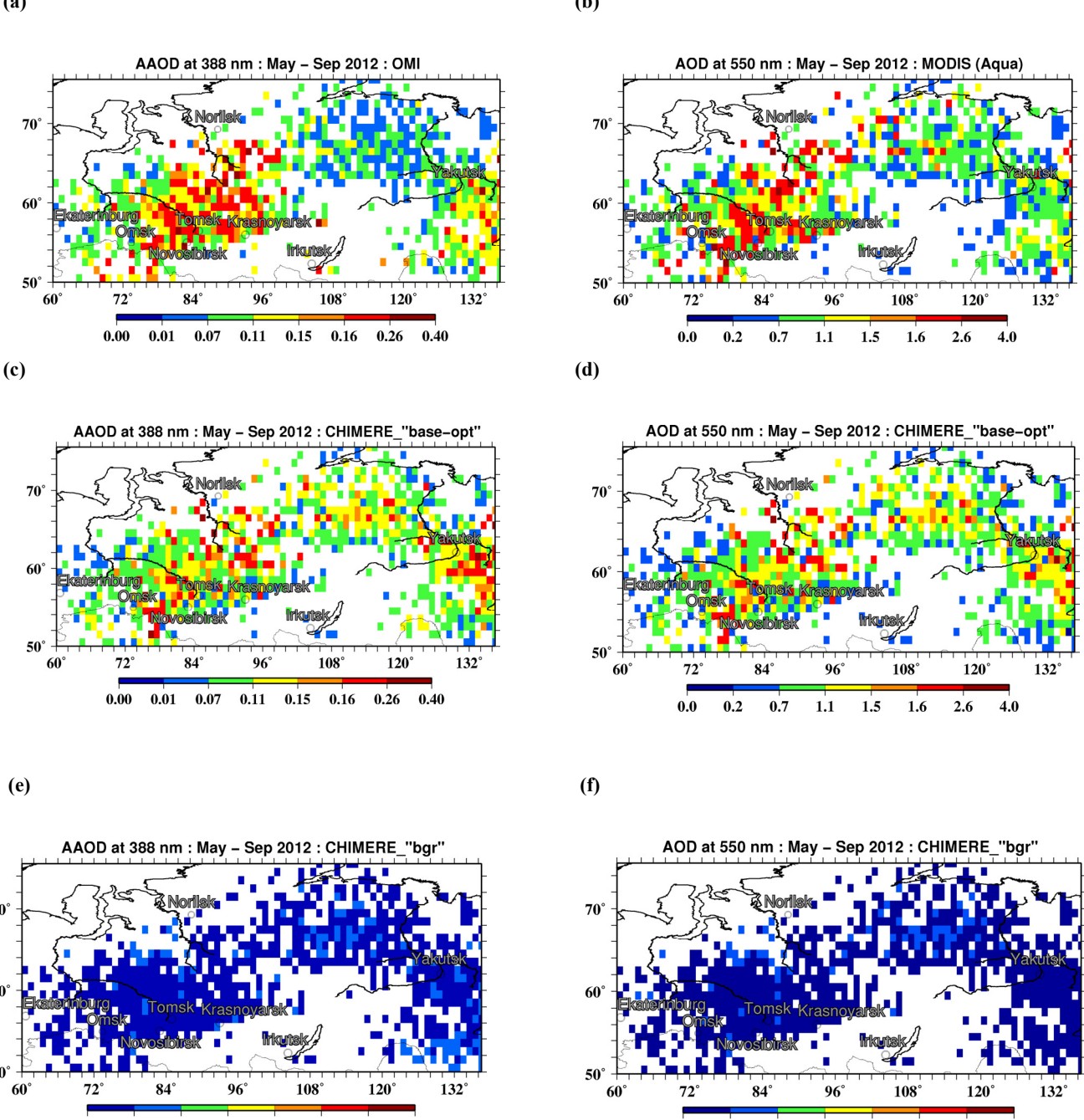

**Figure 5: Spatial distributions of the mean values of AAOD at 388 nm (a, c, e) and AOD at 550 nm (b, d, f) in the period from 1 May to 30 September 2012 according to (a, b) the OMI and MODIS observations, respectively, and simulations performed with the optimized BB emissions (c, d) and without BB emissions (e, f).**

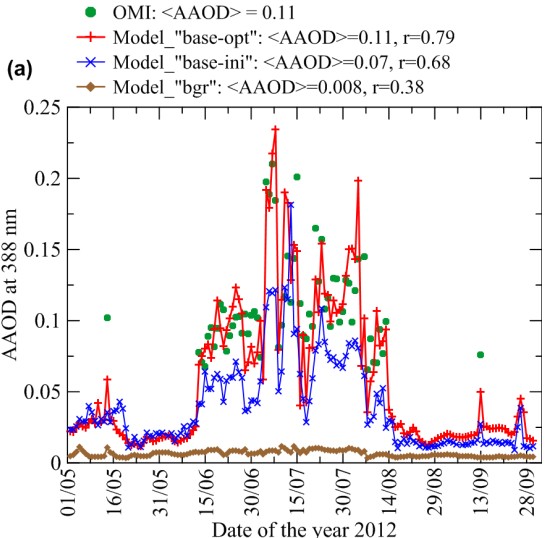
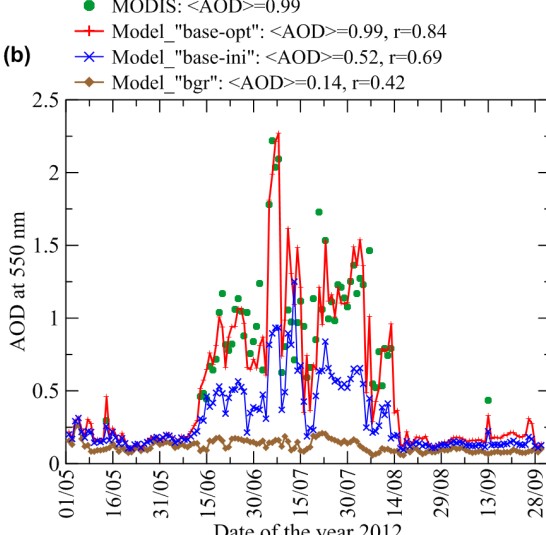

**Figure 6: Time series of daily AAOD (a) and AOD (b) values averaged over the study region according to the OMI AAOD and MODIS AOD observations and simulations ("base-opt" and "base-ini) performed with the optimized and initial-guess BB BC and OC emissions, as well as ("bgr") without fire emissions. Note that whenever a sufficient number of the observational data points is available (see Sect. 4.2), the simulations have been averaged over the same grid cells as the observations; otherwise, the simulated AAOD and AOD values have been averaged over the whole study region. The numbers given in the figure legends report the mean AAOD and AOD values obtained by averaging over the observational data points and their simulated matchups shown in the figure as well as the values of the correlation coefficient.**

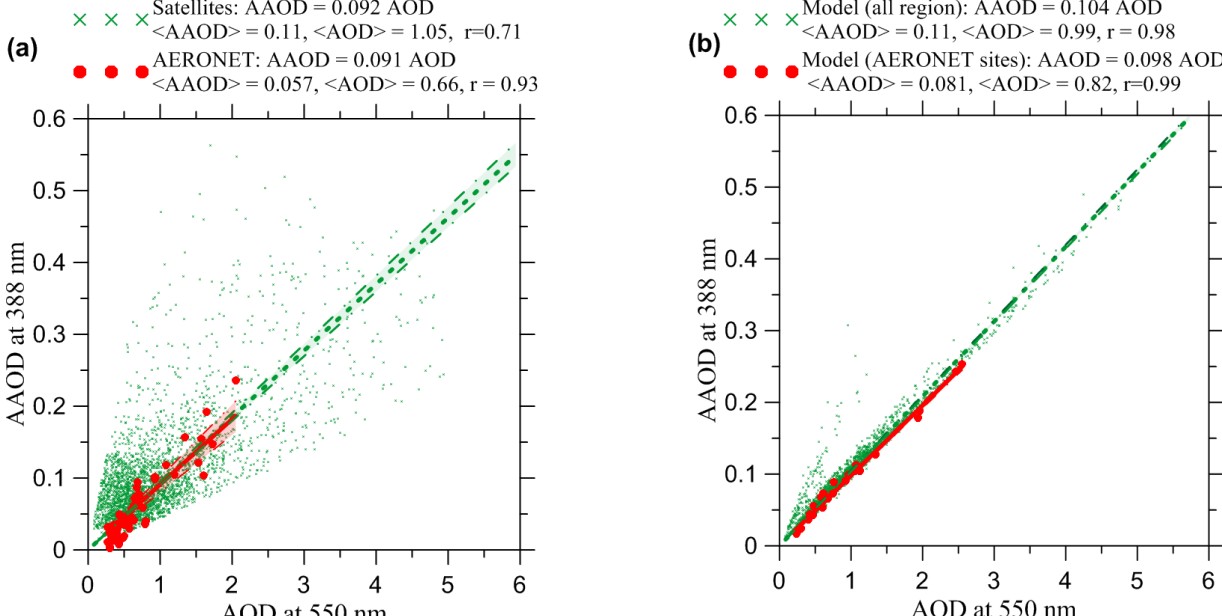

**Figure 7:** The relationship between AAOD and AOD values according to (a) the satellite and AERONET data and (b) corresponding simulated data from the "base-opt" model run. In the case of the satellite data, each data point represents a value of AAOD or AOD for a given cell of the model grid. The AERONET data are described in Sect. 2.4; the corresponding modeled data have been extracted for grid cells and days matching the AERONET observations. The figure legends report the equations of a linear regression without intercept, the mean values of the AAOD and AOD for the different datasets, and the values of the correlation coefficient for each set.

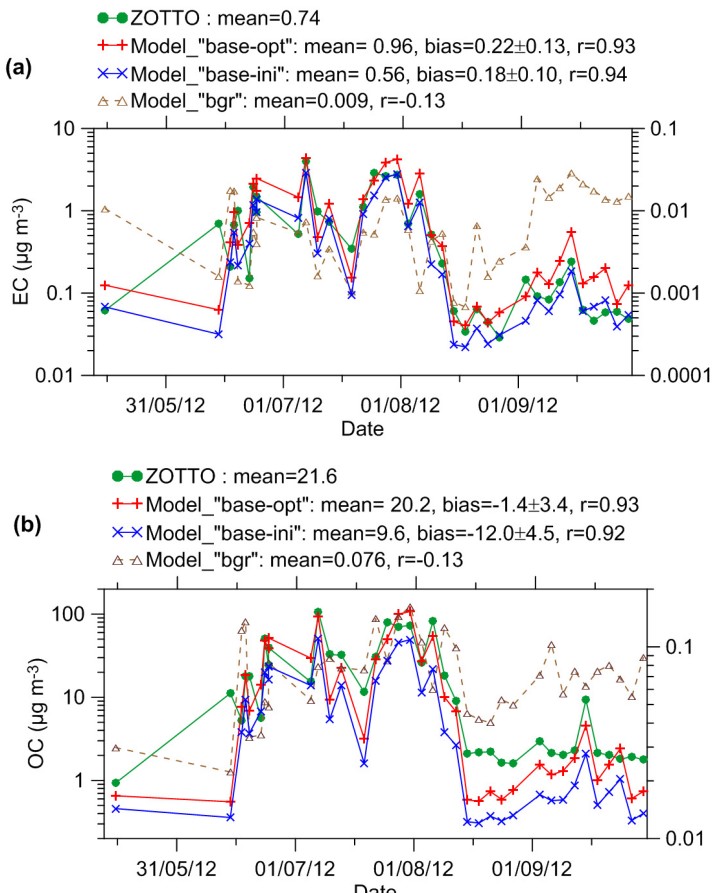

**Figure 8: Time series of the EC and OC mass concentrations (µg m⁻³) measured at ZOTTO in comparison with their simulated matchups from the "base-opt", "base-ini" and "bgr" simulations. Values of several statistical characteristics are reported in the figure legends; the confidence intervals for the biases are evaluated in terms of the 90th percentile. The EC and OC concentrations from the "bgr" simulation are plotted using the corresponding axes on the right-hand side of the panels.**

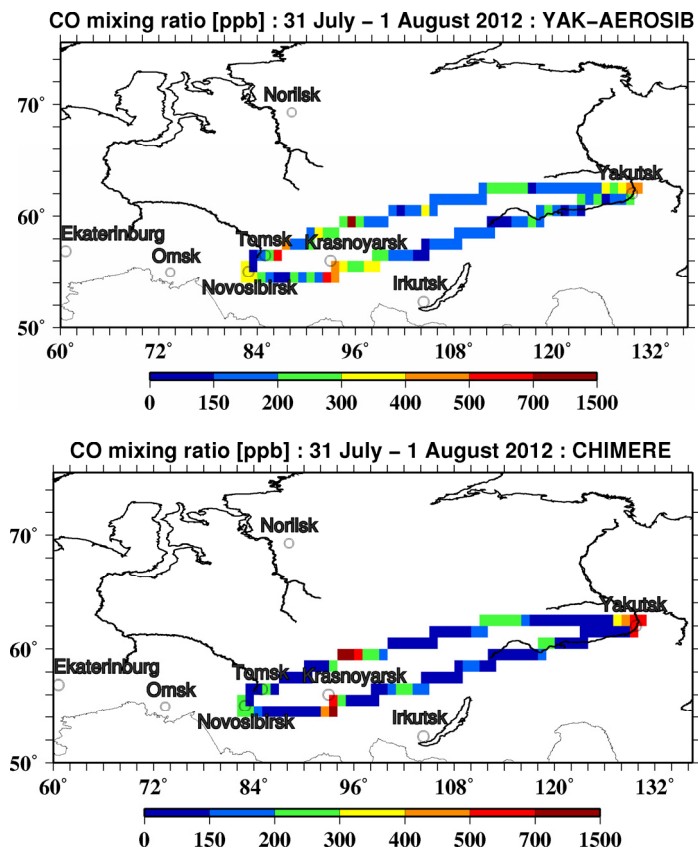

**Figure 9: CO mixing ratios (ppb) derived from the measurements that were made in the framework of the YAK-AEROSIB experiment on 31 July and 1 August 2012 (a) in comparison with corresponding simulated values from the "base-opt" CHIMERE run (b). The mixing ratios are overlaid on the CHIMERE grid: their values have been calculated by averaging the original measurement data and their simulated matchups over the region covered by each grid cell that was intersected by the aircraft trajectory.**

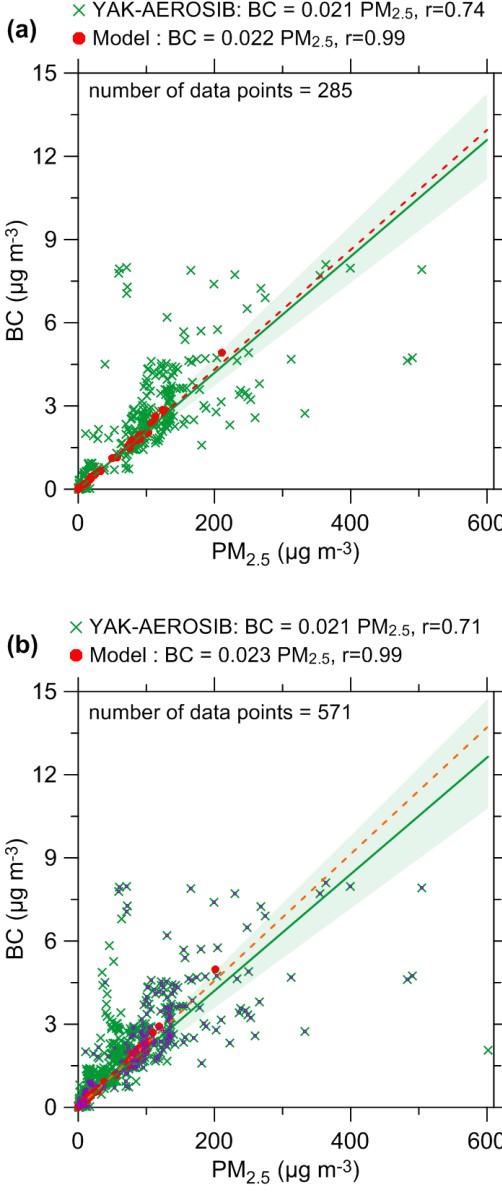

**Figure 10: Relationships between BB BC and PM$_{2.5}$ concentrations obtained from the YAK-AEROSIB observations and from the simulations performed with the optimized BC and OC emissions. The relationships have been obtained by selecting PM$_{2.5}$ and BC concentrations matching (in space and time) the measured CO mixing ratios that exceed the 90$^{th}$ percentile (a) or 80$^{th}$ percentile (b) of the observed distribution of the CO mixing ratios. Purple dots depicted in the panel (b) represent the observational data shown in the panel (a). The figure legends give the equations for a simple linear regression without an intercept. The shaded areas indicate the 90$^{th}$ percentile confidence intervals for the linear regression lines fitted to the measurement data.**

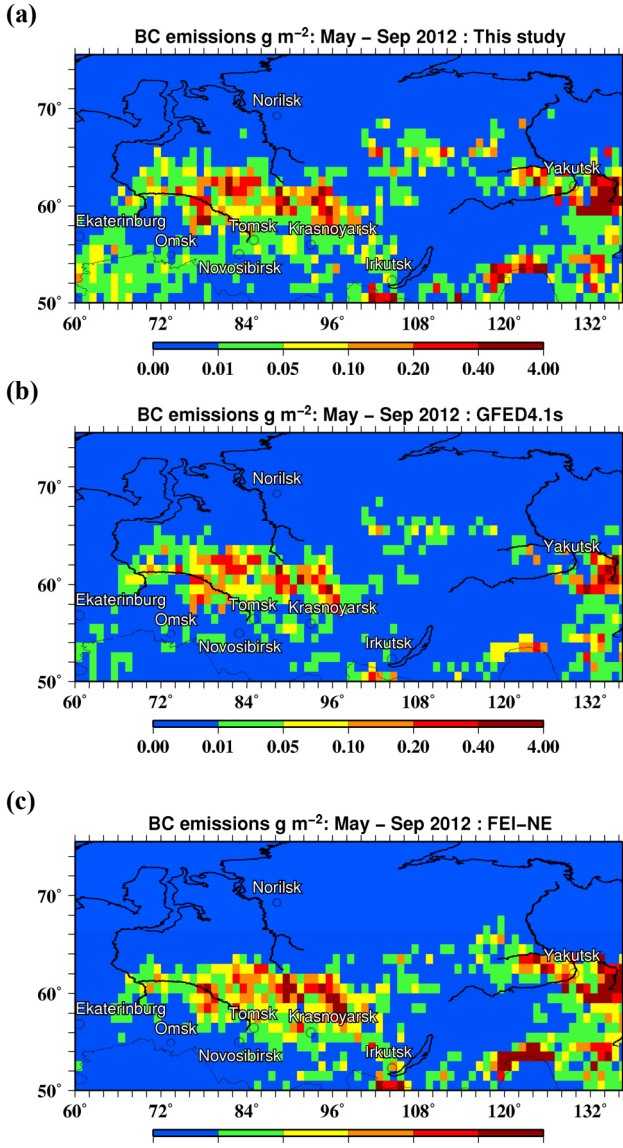

**Figure 11: Gridded estimates of the BB BC emission totals (g m⁻²) obtained in this study (a) and calculated using GFED4 (b) and FEI-NE (c) data for the period from 1 May to 30 September 2012.**

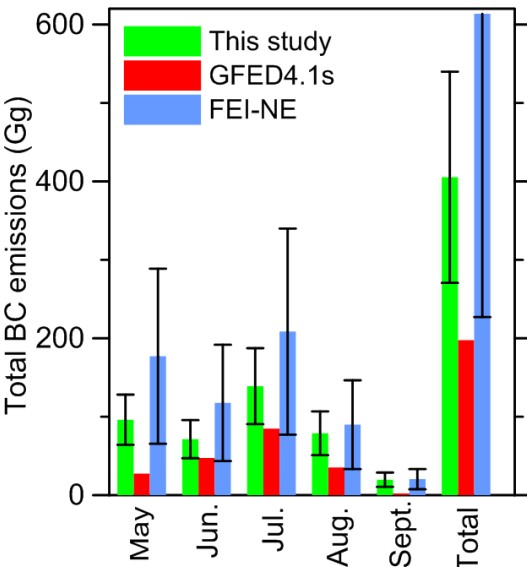

**Figure 12: BC amounts (Gg) emitted from fires in the study region in May-September 2012: the estimates constrained by satellite AAOD and AOD satellite observations are presented in comparison to corresponding estimates calculated with the GFED4.1s and FEI-NE data. The error bar in the positive direction for the FEI-NE estimate is not shown to improve readability of the figure.**

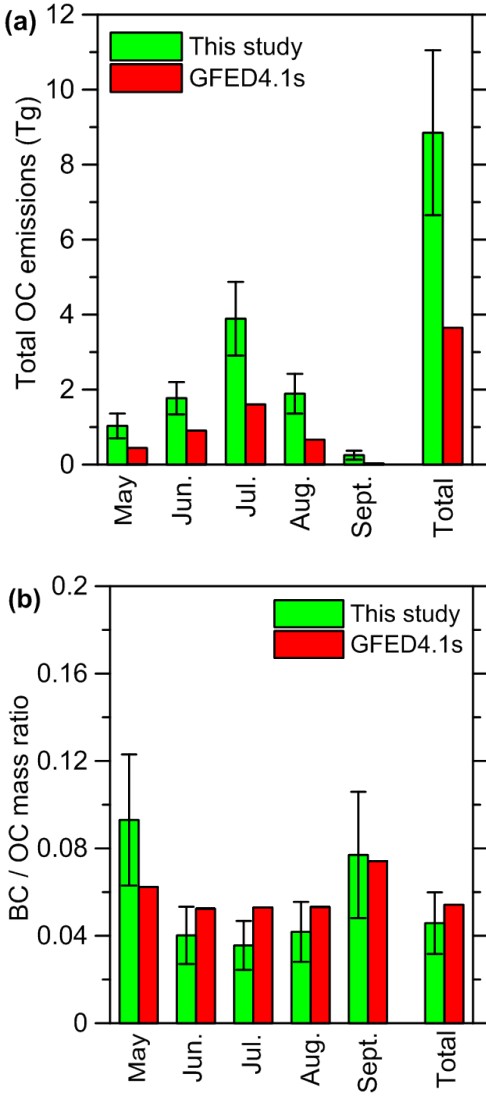

**Figure 13: (a) OC amounts (Tg) emitted in the study region in May-September 2012 according to this study and the GFED4 data along with (b) the corresponding estimates of the BC/OC emission ratios (g g$^{-1}$).**

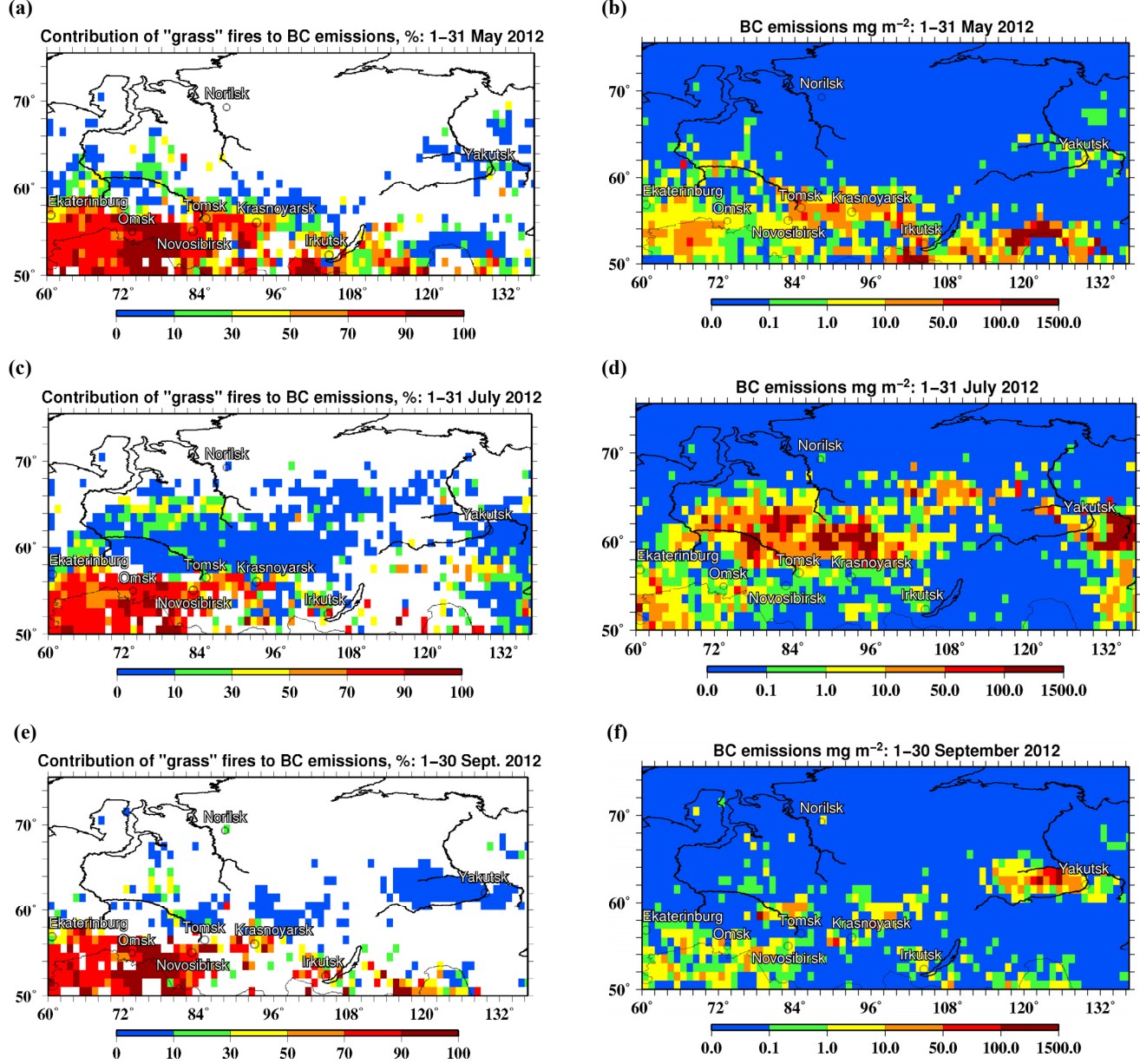

**Figure 14: Spatial distributions of the relative contribution of both grassland and agricultural ("grass") fires to the BB BC emissions integrated over a monthly period (a,c,d) along with spatial distributions of the corresponding BB BC emission values (b,d,f) for May (a,b), July (c,d), and September (e,f), 2012. The distributions were obtained using Eq. (2) and the optimal estimates of the correction factors, $F^{BC}$.**

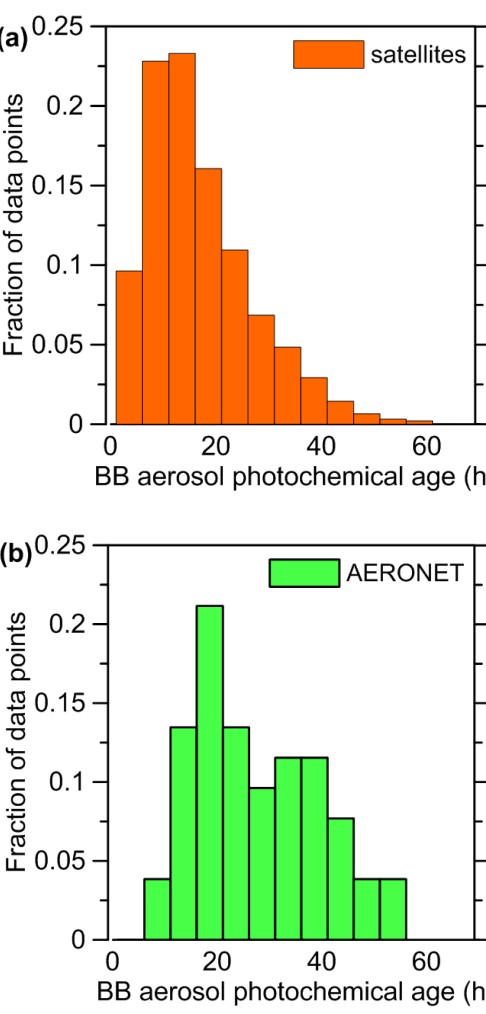

**Figure 15: Histograms of the BB aerosol photochemical ages estimated in accordance with Eq. (1) for the (a) satellite and (b) AERONET data selected for this study. Note that very minor fractions of the data points, which correspond to the age exceeding 60 h and 56 h in the cases of satellite and AERONET data, respectively, are not represented in the histograms for the sake of their better readability.**

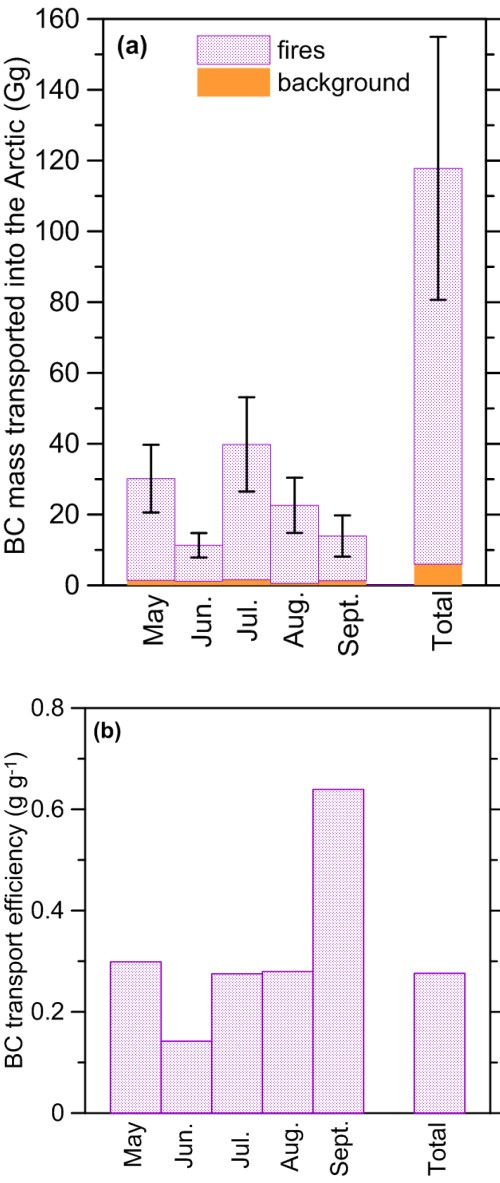

**Figure 16: (a) The estimates of the BC mass (Gg) transported from the study region across the polar circle into the Arctic along with (b) the corresponding estimates of the BC transport efficiency (see Sect. 3.7).**