# Peer review of "Estimation of black carbon emissions from Siberian fires using satellite observations of absorption and extinction optical depths"

_Atmospheric Chemistry and Physics, 2018_

## Referee Comment (RC1) · Anonymous Referee #1 · 8 Jun 2018

**Review of "Estimation of black carbon emissions from Siberian fires using satellite observations of absorption and extinction optical depths"**

**By Konovalov et al. (ACPD 2018)**

This manuscript reports on the challenging topic of estimating black carbon emissions due to biomass burning (BB) and its transport into the Arctic, evaluated during the 2012 fire season in Siberia. The authors comment that this topic is particularly relevant because of the implications of black carbon emissions for Arctic climate, while at the same time it is very challenging because direct observations of black carbon aerosol from fires are sparse, in this vast, remote, region. Indirect methods are developed by the authors, combining in novel ways OMI and MODIS satellite observations of (absorbing) aerosol optical depth, together with in-situ observations of aerosol concentrations and optical depths, as well as modeling.

Through combination of AOD and AAOD satellite observations, the authors propose to constrain the modeled elemental and organic carbon concentrations, and in turn their emissions. Because the model AAOD, directly resulting from CHIMERE, was unreliable, the authors use an empirical relationship between AAOD and AOD, as derived from particular AERONET observations of aerosol that have been attributed to BB. Due to the complexity of the study, the authors are careful in deriving their methodology, and describing the uncertainties involved, both from the modeling perspective and from uncertainties associated to the observations. The uncertainty analysis has been further expanded by performing several sensitivity experiments to investigate the contribution of specific sensitivities to the model results.

Hence, I believe this study can be considered as very thorough, providing top-down constraints on BC emissions from BB. These turn out significantly larger than estimated with GFEDv4, which is often used as a reference. The manuscript is also very well written, hence I recommend it for publication after a few minor questions have been addressed.

- Sensitivity to SOA: Authors discuss this subject as a potential uncertainty, although they write that it is not represented in CHIMERE (pp11, line 10). Also authors write that uncertainties in SOA could affect the optimal estimates of $F^{OC}$ (pp 20, l16), they suggest it does not affect BC as long as 'simulated AOD values are fitted to the AOD observations'. Finally, on pp 27 l14 the authors suggest that this uncertainty could explain some of their differences in OC emissions compared to GFED. I am still not fully satisfied about uncertainties due to this aspect. It is well known that (biomass burning) SOA budgets are poorly constrained (e.g. Spracklen et al., ACP 2011). By not adequately representing them in CHIMERE there is at least uncertainty in dependency of the EC/OC ratio depending on the lifetime of the plume, with fresh plumes, with comparatively little SOA contribution to OC. Can authors please expand on this aspect a bit more? It would be very interesting if the authors are able to test this uncertainty through an actual sensitivity experiment where SOA contribution to aged [OC] would be enhanced.

- Optimization: The authors compute a single monthly mean optimization factors F for the complete region. They find rather different values per month (Table 2). In part this appears to be associated to different type of fires, particularly for grass land compared to boreal forest fires (Fig. 14). I wonder if the method couldn't easily be expanded to optimize these scaling factors but for two sub-regions (e.g. below/above 57N), such that they can better be associated to particular fire types. It would be interesting to see if the scaling factors would then show a more homogeneous value for different months.

**Technical comments:**

-pp11, l2: "Usually"-> "As usual"

- Konovalov et al. (2017a): Please check and complete reference details

-pp21, l17: "Figure 4"-> "Figure 5"

-pp22, l18: "an artifact of": suggest to change into "enhanced due to"

-pp24, l18 "also" (remove)

-pp26, l 29 "their": to what does this refer to? GFED?

-pp29, l24: "predicated"->"predicted"

-pp30, l26 "the mean AAOD": consider to change to "the mean observed AAOD"

-pp32, l1 "are" (remove)

---

## Referee Comment (RC2) · Anonymous Referee #2 · 12 Jul 2018

The main goal of the manuscript by Kanovolov et al. is to investigate the feasibility of constraining biomass burning (BB) BC emissions by using OMI measurements of aerosol absorption optical depth (AAOD). A long-standing difficulty with using OMI AAOD for this goal is that brown carbon can impact the measurements in ways that are difficult to account for within the inverse modeling process. In this work, a combination of relationships between EC, OC, single scattering albedo and AOD measurements are used to cleverly circumvent this traditional shortcoming. The work is thus novel and interesting in this regard, methodologically. This technique is then used to estimate BB emissions in the arctic, providing some valuable top-down estimates of emissions in this region that are significantly different than widely used inventories (e.g., GFED4).

[Figure]

The manuscript is overall fairly well written, although there are some aspects of the inverse modeling procedure that could be clarified. It would also further strengthen the paper if the authors more clearly evaluated the impact of their top-down estimates as compared to simulations without any optimization of emissions scaling factors. These comments and other smaller issues and questions are described below. Addressing these likely amounts to minor revisions, after which this paper will be suitable for publication in ACP.

Comments:

Major:

5.7: I'm not sure that not making us of a priori information is touted as being a feature, nor that this statement is strictly correct. Bayesian inference is superior for a reason, in that the use of prior knowledge is essential to the solution of inverse problems. Further, there are plenty of sources of a priori information used in the methods described here. CHIMERE uses a prior anthropogenic inventory (HTAP) which helps define background values that contribute to the inversion. In another sense, the FRP data serves as a source of prior information to build a BB emission inventory following Konovalov 2014. Further, one could think of the land cover information used therein as being counted as prior information, etc.

11.10: So BC aging not included in CHIMERE? I can understand neglecting SOA, but BC aging affects its lifetime and transport, particularly to the arctic, as shown in previous studies. Accounting for aging should go beyond keeping track of particle age, as a threshold, as aging changes the particle properties and susceptibility to scavenging and removal. Comment on how this assumption biases the results shown here.

Eq 4: I wonder why the authors took this approach rather than relating AAOD/AOD to SSA (which seems a bit more direct, given that SSA is measured, whereas EC/(OC+EC) is estimated from Eq 3 and they have the problem of random errors in the

independent variable so have to use ODR instead of least squares). Then they could use Eq 3 to substitute SSA for the ratio of EC/(EC+OC), and derive an equation similar to Eq 6 that allows for estimation of AAOD without using a model-calculated SSA for BB organic aerosol.

16.4: This is starting to sound a bit circular. Use of CHIMERE here to separate the background AAOD and AOD would imply that CHIMERE has the correct BB emissions. How much do these estimates change if the authors were to run CHIMERE using their posterior BB emissions estimates? Does this entire procedure need to be iterated? Or, by "background", do the authors mean non-BB AOD and AAOD? This would make a little more sense, being as they stated earlier that they don't use CHIMERE for computing AOD and AAOD of BB aerosol given uncertainties in refractive index of BB OC. Still.. It took me several times reading this section and section 2.1 to piece together the process being used. It could benefit significantly from a more clear explanation, that more clearly maps to the schematic in Fig 1. At present, Fig 1 is misleading, as only part (the non BB part) of the AOD and AAOD estimated from CHIMERE are used for calculation of the AAOD value that ultimately is used within the estimation procedure in red. So this part of the figure needs more detail. The source of the non-BB aerosols (HTAP) is another input to the modeling that should be indicated.

Eq 7: Minimization of the least absolute value of the model error corresponds (in a Bayesian sense) to assuming that the observation errors follow a Laplacian distribution. This the of L1 minimization is best applied when the data is suspected of having outliers. The authors state on p 18 that they suspect multiplicative errors. This would most likely result in a log-normal distribution, if the errors being multiplied are Gaussian, rather than a Laplacian distribution. Thus it would make more sense to perhaps optimize the log of the model error. This is, in some sense, what the authors have done by dividing the model error by the observations so that they consider only the relative model error. But the form is not quite the same. Instead, the authors should normalize by the observation error covariance. Not all of the observed AAOD and AOD measurements have the same errors, and thus shouldn't be considered equally in the objective function being minimized here.

18.12: I agree the satellite observations help indicate the location of the fires. But there is still considerable uncertainty in the variability of emissions factors from one fire location to another, especially given the crude representation of land cover types (considering only 5 classes). This sort of uncertainty could lead to large errors in the spatial distribution of the magnitude of BC emissions, even if the fire locations and intensity are perfectly known. Thus, the authors are going to need to admit here that the aggregation error is an unavoidable consequence of the chosen computational method.

12.21: Later in the paper (Figs 5, 6), I was struggling to clarify what constituted the "base" simulation. Is this a simulation using initial estimates of BB emissions, or an estimate based on the application of the scaling factors optimized from Eq 7? It would be useful to specify here. In presentation of figures of simulations that use the optimized emissions, a clearer name (e.g., "optimized") would be better.

Figs 5, 6: As mentioned above, not clear what is "base" here. Simulation with scaling factors of 1.0? Or the "base case option" (p18, line 25) of the optimized results? Please clarify.

Fig 6: Either way, it would be instructive to see Fig 6 with the emissions scaling factors of 1.0 as well as with the optimized emissions scaling factors, to see the improvement owing to the inversion.

Minor:

Intro: Missing some papers on BC transport to the arctic, of which a few more have been recently published.

3.9-11: References on use of atmospheric observations to constrain BB emissions is a bit thin. There have been numerous studies in this area. Even some of the BB

emission products use ambient measurements as part of their constraints, such as QFED. Update: some such studies are referenced later in the introduction; still it seems like something is missing at this point.

4.16-19: And also dust? Perhaps that's not a problem when focused on regions suspected of being influenced by BB, but the statement as currently written is more general than that. Also, aerosol layer height is an important source of uncertainty in these types of studies.

11.31: I'm confused by this statement. It seems the previous content was all about describing how AAOD is calculated in CHIMERE. But here the authors state that AAOD values from CHIMERE that include BB emissions are not used? This seems to contradict page 10, line 21 "we took into account BB emissions . . ." UPDATE: after reading the entire paper, this becomes clear. But it could perhaps be written here in a way that doesn't lead to an initial source of confusion.

18.7: Or it would require note more computational resources, but more computational tools, such as an adjoint or other (i.e. Lagrangian) source-oriented model. CHIMERE has an adjoint (Menut et al., 2000) although perhaps it is antiquated.

Fig 4: Does the non-zero intercept of this line tell you something about the refractive index of the OC?

Fig 7(b): I'm not sure this shows much other than the fact that the modeled EC/(EC+OC) ratio is rather consistent, which is likely owing the fact that the model is based on a fixed EC/OC ratios in the emissions. Still, the authors should comment on the one feature that runs counter to this trend in the figure, the cluster of green points above the 1:1 line, which potentially indicates sampling of regions influenced by non-BB sources.

23.7: The authors need to clarify the writing here with regards to precision vs accuracy. If the errors were truly random, they would not lead to bias.

Fig 8: The way the background concentrations are included has the downside of skewing the vertical scale such that differences between the model and observations appear minimized, visually. Instead, a linear scale should be used, and background concentrations could be plotted using a different vertical axis on the right hand side, or scaled by x10-100 so they could be plotted on the same scale. Also, more interesting than showing the background concentrations would be to show the model estimates when using the BB emissions with the optimized vs unoptimized emissions factors. Since "base" wasn't clearly defined in the presentation of the results, it's not clear to me which of these cases is being presented anyways.

Fig 10: The two cases don't appear to be significantly different; I would suggest just showing one. Also, I don't understand what the purple dots in panel (b) represent.

24.27: I thought it was stated in the CHIMERE model description that there was no aging of BC included in the model.

26.7: While uncertainty information isn't provided with the GFED4 inventory, there have been numerous studies evaluating this inventory compared to other bottom-up inventories as well as evaluations based on top-down constraints using ambient measurements. These studies provide some range of estimates for the uncertainty in GFED4 emissions that could be referenced. In other words, from previous work we know that it's not something small like 1%, and certainly error as large as x2 (or even x10, regionally) have been noted.

Editorial:

4.5 and other locations: Careful with random switches between present and past tense.

4.10: GEOS-Chem

8.12: "measurement data" is a little redundant.

10.32: distributions –> distribution

11.4: are anyway not –> are not

17.5: The authors cite Enting 2001, but the bibliography only includes Enting 2002.

Fig 2 and 9: Text in the figure is illegible. Suggest using solid black, font that is easier to read.

Fig 5: Small suggestion on labeling: Don't write "CHIMERE - bgr" or "CHIMERE - base", as that at first looks like subtraction. Just leave out the hyphen, or use a colon.

21.17: Figure 5?
* * *

---

## Author Comment (AC1) · 18 Sep 2018

We are very grateful to the Referee for the positive evaluation of our manuscript and for the useful comments and suggestions. Below we describe our point-by-point responses to the Referee's comments.

Referee's comment: *Sensitivity to SOA: Authors discuss this subject as a potential uncertainty, although they write that it is not represented in CHIMERE (pp11, line 10). Also authors write that uncertainties in SOA could affect the optimal estimates of F∧OC (pp 20, l16), they suggest it does not affect BC as long as 'simulated AOD values are fitted to the AOD observations'. Finally, on pp 27 l14 the authors suggest that this un-*

*certainty could explain some of their differences in OC emissions compared to GFED. I am still not fully satisfied about uncertainties due to this aspect. It is well known that (biomass burning) SOA budgets are poorly constrained (e.g. Spracklen et al., ACP 2011). By not adequately representing them in CHIMERE there is at least uncertainty in dependency of the EC/OC ratio depending on the lifetime of the plume, with fresh plumes, with comparatively little SOA contribution to OC. Can authors please expand on this aspect a bit more? It would be very interesting if the authors are able to test this uncertainty through an actual sensitivity experiment where SOA contribution to aged [OC] would be enhanced.*

Indeed, the uncertainty in the SOA budget affects the EC/OC ratio and its dependence on the BB aerosol age. However, the corresponding errors in our BC emission estimates are mostly compensated by similar errors in the simulated AOD values. The discussion of this point is expanded in the revised manuscript.

In particular, the compensation of possible errors in the EC/OC ratio and AOD is demonstrated (see Sect. 2.2.4 and Eq. 11) by simplifying the original empirical relationship (given by Eq. 6) between the BC and OC column, AOD and AAOD. Furthermore, as suggested by the Referee, we performed a sensitivity experiment, in which the yields of all of the SOA species from oxidation of the major volatile SOA precursors (such as toluene, xylenes, isoprene and terpenes) were enhanced by a factor of 7 with respect to the base case, whereas the reaction list and the reaction rates (as well as all other simulation settings) were kept the same. As a result of this model modification, the relative enhancement of the averaged (over the whole period and region considered) particulate organic matter (POM) column amounts due to SOA formation increased from only 2.6 % (in the base case simulation with the optimized emissions) up to 27 % (in the test case). The increased SOA contribution to POM in our test case simulation corresponds to the upper bound of the wide range of the BB POM enhancements observed in aging BB plumes in several field studies in North America. This numerical experiment is discussed in Sect. 3.6 of the revised manuscript as the

test case No. 2. We found that the optimal BC emission estimate changed rather insignificantly, having increased only by $\sim$ 5 %. It is not surprising, however, that our estimates of the OC emissions turned out to be much more sensitive to the SOA formation; specifically, the top-down estimate of the total OC emissions dropped by $\sim$15 % in the test case as a result of an increase in the BB aerosol abundances and a slight decrease in the mass extinction efficiency. We would like to note that, to the best of our knowledge, any aerosol modeling scheme which could ensure quite adequate simulations of the evolution of the organic fraction of BB aerosol in Siberia or elsewhere is still not available: presently, the elaboration of such a scheme is an active research problem.

Referee's comment: *Optimization: The authors compute a single monthly mean optimization factors F for the complete region. They find rather different values per month (Table 2). In part this appears to be associated to different type of fires, particularly for grass land compared to boreal forest fires (Fig. 14). I wonder if the method couldn't easily be expanded to optimize these scaling factors but for two sub-regions (e.g. below/above 57N), such that they can better be associated to particular fire types. It would be interesting to see if the scaling factors would then show a more homogeneous value for different months.*

Unfortunately, the optimization method employed in this paper could not be easily expanded to optimize the scaling factors for two (or more) sub-regions: we would need to use a different cost function, a different optimization procedure and perform more model runs. However, if a sub-region is sufficiently large, aerosol transport between the sub-regions can be disregarded, and the correction factors for the BB emissions in a given sub-region can be evaluated by using the same optimization method.

Accordingly, we performed an additional analysis using the AAOD and AOD observations only over a selected sub-region (50-57N, 60-115E) where the relative contribution of agricultural and grass fires to FRP was much larger and more uniformly distributed across the different months than in the whole region. The estimates of the $F^{BC}/F^{OC}$ ratio obtained for this sub-region show much smaller variations between different months, as compared to the corresponding estimates for the whole region, whereas monthly variations of the $F^{BC}$ and $F^{OC}$ factors themselves even increased (which may be due to monthly variations of the conversion factor). Therefore, our additional analysis supports (although does not prove) the possibility that the monthly variations of the $F^{BC}/F^{OC}$ ratio are associated with different fire types and thus indicates that that the emission factors specified in the GFED4 inventory and in our simulation for either BC or OC or both are biased in case of at least one fire type. The analysis is described in Sect. 3.1 of the revised manuscript and in a new section (Sect. S3) of the Supplementary material.

Following the Referee's suggestion, we also performed a similar analysis for a sub-region above 57N. However, we found that a contribution of agricultural and grass fires to the integral FRP in May in this sub-region (32 %) was not much smaller than that in the entire study region (41 %). For this reason, the analysis of the second sub-region could not provide any conclusive results in regard to the question raised by the Referee and thus is not presented in the revised manuscript.

Referee's comments: *Technical comments:*

*-pp11, l2: "Usually"-> "As usual"*

*- Konovalov et al. (2017a): Please check and complete reference details*

*-pp21, l17: "Figure 4"-> "Figure 5"*

*-pp22, l18: "an artifact of": suggest to change into "enhanced due to"*

*-pp24, l18 "also" (remove)*

*-pp26, l 29 "their": to what does this refer to? GFED?*

*-pp29, l24: "predicated"->"predicted"*

*-pp30, l26 "the mean AAOD": consider to change to "the mean observed AAOD"*

[Figure]

*-pp32, l1 "are" (remove)*

We thank the Referee for these technical comments. All the suggested changes and corrections are introduced in the revised manuscript.

---

## Author Comment (AC2) · 18 Sep 2018

We thank the Referee for the overall positive evaluation of our manuscript and for the thoughtful comments. We have carefully revised the manuscript according to the Referee's suggestions. Our point-by-point responses to the Referee's specific comments are provided below.

Referee's comment: *5.7: I'm not sure that not making use of a priori information is touted as being a feature, nor that this statement is strictly correct. Bayesian inference is superior for a reason, in that the use of prior knowledge is essential to the solution of inverse problems. Further, there are plenty of sources of a priori infor-*

[Figure]

*mation used in the methods described here. CHIMERE uses a prior anthropogenic inventory (HTAP) which helps define background values that contribute to the inversion. In another sense, the FRP data serves as a source of prior information to build a BB emission inventory following Konovalov 2014. Further, one could think of the land cover information used therein as being counted as prior information, etc.*

We are very sorry if one of our introductory remarks looked like "touting": our goal was simply to provide a reader with some tentative idea about our method. Certainly, using the Bayesian inference has, in a general case, serious advantages. However, not using it in the particular case addressed in our study allowed us to avoid any subjective quantitative assumptions regarding the model and observation errors and the uncertainties in a priori emission data. Such assumptions could otherwise affect our a posteriori BC emission estimates and their confidence intervals. We agree also that our study uses many sources of a priori information, even though they were not used in a Bayesian sense (as was mentioned in the reviewed manuscript). To avoid possible misunderstanding, the statements criticized by Referee are removed from the introduction of the revised manuscript. Our method is described in detail in Sect. 2.2.4.

Referee's comment: *11.10: So BC aging not included in CHIMERE? I can understand neglecting SOA, but BC aging affects its lifetime and transport, particularly to the arctic, as shown in previous studies. Accounting for aging should go beyond keeping track of particle age, as a threshold, as aging changes the particle properties and susceptibility to scavenging and removal. Comment on how this assumption biases the results shown here.*

We thank the Referee for this insightful comment. We recognize that the discussion of this point was not sufficient in the reviewed manuscript. Indeed, any aging changes in susceptibility of aerosol particles for scavenging and removal were not taken into account in our simulations. Presently, the susceptibility of particles for in-cloud scavenging (and for subsequent wet deposition) is described in CHIMERE by the constant empirical uptake coefficient, which was set to be zero in our simulations of the BB

aerosol and was set to be unity in the simulation of the background aerosol fraction. That is, BB aerosol was assumed in our simulations to be hydrophobic in regard to in-cloud scavenging. This assumption could result in some underestimation of our BB BC emission estimates, as aerosol aging processes are known to increase hygroscopicity of BC-containing aerosol particles and to accelerate their removal from the atmosphere by precipitation through scavenging.

To address this comment of the Referee, we performed an additional sensitivity test (presented in Sect. 3.6 of the revised manuscript as the Test case No. 3), in which the empirical uptake coefficient was set to be unity. The use of this simulation instead of the base case simulation with the optimized emissions in our estimation procedure resulted only in a minor increase ($\sim$6 %) in our optimal estimate of the total BB BC emissions; the changes in the monthly BC emission estimates were found to be similarly small. Such a result is hardly surprising because the major fires considered in our analysis occurred mostly during dry periods with low precipitation. The same test simulation was used to assess the effect of aerosol scavenging on our computations of BC amounts transported into the Arctic. We found (as indicated in Sect. 3.7 of the revised manuscript) that the overall transport efficiency evaluated under the assumption that BB aerosol particles are composed of hydrophilic material turned out to be only slightly smaller (25.2 %) than the corresponding base case estimate (27.6 %). Note that taking into account that BC constitutes typically a very minor fraction of BB aerosol particles in Siberia and that BB BC is usually covered by organic material, we preferred to avoid using the notion "BC aging", which is usually used to characterize changes in properties of BC-containing aerosol from fossil fuel burning. Instead, we used a more general notion "BB aerosol aging".

Referee's comment: *Eq 4: I wonder why the authors took this approach rather than relating AAOD/AOD to SSA (which seems a bit more direct, given that SSA is measured, whereas EC/(OC+EC) is estimated from Eq 3 and they have the problem of random errors in the independent variable so have to use ODR instead of least squares). Then*

*they could use Eq 3 to substitute SSA for the ratio of EC/(EC+OC), and derive an equa-tion similar to Eq 6 that allows for estimation of AAOD without using a model-calculated SSA for BB organic aerosol.*

Indeed, SSA is retrieved from the OMI measurements but the SSA retrievals are much less abundant than the AAOD retrievals. This was explained in Sect. 2.1.1 of the reviewed manuscript ("However, compared to AOD and SSA retrievals, AAOD is less affected by cloud contamination due to the partial cancellation of errors in AOD and SSA. OMAERUV reports AOD/SSA/AAOD with the associated quality flags '0' and '1'. While all three retrievals are reliable with the quality flag '0', only AAOD is reliable with either quality flag. Accordingly, the number of reliable AAOD retrievals is greater than the number of those of AOD and SSA."). Therefore, we opted for using the OMI AAOD retrievals in combination with the MODIS AOD retrievals, instead of using the direct retrievals of SSA. An additional explanatory remark is introduced in Sect.2.2.3 of the revised manuscript. Note that even if we used directly the satellite observations of SSA, we would still have a problem of random errors in the independent variable when relating the AERONET SSA measurements at two different wavelengths.

Referee's comment: *16.4: This is starting to sound a bit circular. Use of CHIMERE here to separate the background AAOD and AOD would imply that CHIMERE has the correct BB emissions. How much do these estimates change if the authors were to run CHIMERE using their posterior BB emissions estimates? Does this entire procedure need to be iterated? Or, by "background", do the authors mean non-BB AOD and AAOD? This would make a little more sense, being as they stated earlier that they don't use CHIMERE for computing AOD and AAOD of BB aerosol given uncertainties in refractive index of BB OC. Still.. It took me several times reading this section and section 2.1 to piece together the process being used. It could benefit significantly from a more clear explanation, that more clearly maps to the schematic in Fig 1. At present, Fig 1 is misleading, as only part (the non BB part) of the AOD and AAOD estimated from CHIMERE are used for calculation of the AAOD value that ultimately is used*

[Figure]

*within the estimation procedure in red. So this part of the figure needs more detail. The source of the non-BB aerosols (HTAP) is another input to the modeling that should be indicated.*

Yes, the "background" AAOD and AOD values are meant to represent the conditions without BB aerosol sources in the study region (as was stated three lines below in the same page). To simplify understanding of this point, we additionally explained in the revised manuscript (directly in the former line 4 of page 16) that the background AAOD and AOD values were predicted in the CHIMERE simulation without BB emissions. Furthermore, we re-composed Figure 1 as suggested by the Referee; the revised figure indicates the HTAP inventory and the simulations for the "background" conditions among the input data for our analysis.

Referee's comment: *Eq 7: Minimization of the least absolute value of the model error corresponds (in a Bayesian sense) to assuming that the observation errors follow a Laplacian distribution. This the of L1 minimization is best applied when the data is suspected of having outliers. The authors state on p 18 that they suspect multiplicative errors. This would most likely result in a log-normal distribution, if the errors being multiplied are Gaussian, rather than a Laplacian distribution. Thus it would make more sense to perhaps optimize the log of the model error. This is, in some sense, what the authors have done by dividing the model error by the observations so that they consider only the relative model error. But the form is not quite the same. Instead, the authors should normalize by the observation error covariance. Not all of the observed AAOD and AOD measurements have the same errors, and thus shouldn't be considered equally in the objective function being minimized here.*

We thank the Referee, especially, for this thoughtful comment. A correct choice of the objective (cost) function is of a paramount importance for any atmospheric inverse modeling study, but this point, unfortunately, rarely receives significant attention in corresponding publications. Due to the usually complex nature of observational and model errors and the fact that the statistical properties of these errors are often poorly known,

it is commonly assumed that the errors are additive and Gaussian. This assumption leads to a least-squares problem (and its regularized versions) and usually results in sufficiently adequate a posteriori emission estimates. However, as mentioned in Sect. 2.2.4, a multiplicative component of the model errors associated with simulations of the BB components in the atmosphere is expected to cause some underestimation of BB emission estimates. On the other hand, we could not assume that the errors in BB fractions of AAOD were truly multiplicative, since we found that whereas the BB fraction of AAOD in the simulations is always positive, its observational matchups obtained by subtracting the background AAOD fraction from the observed AAOD values might occasionally be negative (due to uncertainties in the background aerosol simulations and errors in the observational matchups of the BB AAOD simulations). Therefore, we, unfortunately, could not follow the Referee's suggestion to optimize the emissions by assuming a log-normal distribution for the errors. We would like to note further that we did not actually perform the standard L1 minimization as apparently presumed by the Referee. Instead, as explained in Sect. 2.2.4 of the reviewed manuscript, we minimized "relative differences between the **mean values** of both AOD and AAOD simulated with optimized BB emissions and their pre-selected matchups derived from satellite measurements" in accordance to Eq. (7) (where the absolute values are computed **after** the summation of the differences between the observations and simulations but not **before** it as in the standard L1 minimization). We explained also that "the optimization of $F^{BC}$ and $F^{OC}$ in accordance with Eq. (7) is equivalent to establishing a simple balance between spatially- and temporally-averaged AAOD (or AOD) retrievals and their simulated matchups on a monthly basis". By dividing the absolute difference between the mean values of AAOD (or AOD) from the simulations and observations to the mean observed value, we establish a criterion for the numerical precision of our emission estimates. We agree that not all of the observed AAOD and AOD measurements have the same errors. However, we do not truly know how the errors depend on the corresponding AAOD or AOD values from observations and simulations, and thus we preferred to avoid making any specific assumptions about these dependencies.

To address the Referee's comment, we have substantially simplified Equations (7), (9) and (10), so that a reader could more easily follow the idea of our simple method. We also have expanded the discussion of our optimization criterion by providing an additional section (S2) in the Supplementary material, where we analyze the effect of unknown observation and model errors on our top-down BC emission estimates. We argue, in particular, that given the unlimited number of unbiased simulations and observations representative of an entire (sufficiently large) region, the estimation error can be expected to approach zero (irrespectively of the statistical distributions of the errors). Since the amount of the data is always limited, the overall estimation error depends on the mean values of the observation and model errors; it also includes an aggregation error.

Referee's comment: *18.12: I agree the satellite observations help indicate the location of the fires. But there is still considerable uncertainty in the variability of emissions factors from one fire location to another, especially given the crude representation of land cover types (considering only 5 classes). This sort of uncertainty could lead to large errors in the spatial distribution of the magnitude of BC emissions, even if the fire locations and intensity are perfectly known. Thus, the authors are going to need to admit here that the aggregation error is an unavoidable consequence of the chosen computational method.*

Indeed, due to the crude representation of spatial and temporal variability in the emission and conversion factors involved in our emission model as well as due to uncertainties in the FRP observations, the actual emission fields can be substantially different from those specified in our simulations. This fact is likely to result in an aggregation error (as was noted in the reviewed manuscript). However, unlike in situ observations, which may not be representative of a part of the strong emission sources in a large region, the satellite AAOD and AOD observations cover (in a quasi-uniform manner) all areas where BB emissions were important. For this reason, we do not expect the aggregation error to be large in our case.

The discussion of an aggregation error is extended in Sect. 2.2.4 of the revised manuscript. Furthermore, the aggregation error is also discussed in the revised Supplementary material (see Sect. S2). We argue that the aggregation error is unlikely to exceed the confidence intervals for our estimates, but we admit that we cannot provide a reliable quantitative estimate for it and note that the likely presence of the aggregation error emphasizes the importance of validation of our estimates by using independent observations.

Referee's comments: *12.21: Later in the paper (Figs 5, 6), I was struggling to clarify what constituted the "base" simulation. Is this a simulation using initial estimates of BB emissions, or an estimate based on the application of the scaling factors optimized from Eq 7? It would be useful to specify here. In presentation of figures of simulations that use the optimized emissions, a clearer name (e.g., "optimized") would be better.*

*Figs 5, 6: As mentioned above, not clear what is "base" here. Simulation with scaling factors of 1.0? Or the "base case option" (p18, line 25) of the optimized results? Please clarify.*

We evaluated the simulations with the optimized emissions. We presumed that this was sufficiently clearly explained in the first paragraph of Sect. 3.2. It would hardly be logical to evaluate some arbitrary initial estimates of the BB emissions instead of the optimized ones.

To simplify the understanding of this point, we have split the initial "base" scenario in Sect. 2.2.1 into the "base-opt" and "base-ini" scenarios (with the optimized and unoptimized emissions, respectively). The corresponding scenario labels are used in the legends and/ or captions of Figures 5-9.

Referee's comment: *Fig 6: Either way, it would be instructive to see Fig 6 with the emissions scaling factors of 1.0 as well as with the optimized emissions scaling factors, to see the improvement owing to the inversion.*

The magnitude of BB emissions calculated with the scaling factors of 1.0 depends on the empirical (conversion) factor relating FRP to the rate of biomass burning (see Eq. 2). The assumed value of this factor is based on a laboratory experiment, and it is not necessarily applicable to satellite FRP observations. Further, our "initial" BB emissions depend on the way to process the FRP observations (in particular, we could, in principle, consider daily mean FRP values instead of the daily maximums of the FRP values, and that would result in quite different values of the emissions). Accordingly, our BB emission estimates calculated with the scaling factors of 1.0 should be considered as being rather arbitrary and, as such, cannot be recommended for the use in atmospheric models (unlike typical "bottom-up" emission inventory data traditionally used as "a priori" emission estimates in inverse modeling studies). For this reasons, we did not originally present any results obtained with the scaling factors of 1.0. Nonetheless, following the Referee's recommendation, we have included the corresponding simulated data in Fig. 6 of the revised manuscript.

Referee's comment: *Intro: Missing some papers on BC transport to the Arctic, of which a few more have been recently published.*

We have updated a list of the papers devoted to transport of BC to the Arctic. Please note that we did not intend this list to be exhaustive, as the BC transport to the Arctic is certainly not the main subject of our study.

Referee's comment: *3.9-11: References on use of atmospheric observations to constrain BB emissions is a bit thin. There have been numerous studies in this area. Even some of the BB emission products use ambient measurements as part of their constraints, such as QFED. Update: some such studies are referenced later in the introduction; still it seems like something is missing at this point.*

Indeed, the references on the use of atmospheric observations to constrain BB emissions were meant to appear not in this place but later in the text. However, a few more BB emission inventories, including QFED, are mentioned in this paragraph in the

revised manuscript.

Referee's comment: *4.16-19: And also dust? Perhaps that's not a problem when focused on regions suspected of being influenced by BB, but the statement as currently written is more general than that. Also, aerosol layer height is an important source of uncertainty in these types of studies.*

In the given context, we could not discuss all possible factors which might hamper the use of the OMI AAOD observations for estimation of BC emissions in a general case. To make our statement more accurate, we have changed the beginning of the corresponding sentence from "the main difficulty is" to "one of the main difficulties is".

Referee's comment: *11.31: I'm confused by this statement. It seems the previous content was all about describing how AAOD is calculated in CHIMERE. But here the authors state that AAOD values from CHIMERE that include BB emissions are not used? This seems to contradict page 10, line 21 "we took into account BB emissions . . ." UPDATE: after reading the entire paper, this becomes clear. But it could perhaps be written here in a way that doesn't lead to an initial source of confusion.*

We have clarified in Sect. 2.2.1 of the revised manuscript that we calculated the BB fraction of AAOD by using an empirical parameterization described in Sect. 2.2.3 but that the AAOD values simulated directly with CHIMERE were used to characterize the background atmospheric conditions (in the absence of fires).

Referee's comment: *18.7: Or it would require note more computational resources, but more computational tools, such as an adjoint or other (i.e. Lagrangian) source-oriented model. CHIMERE has an adjoint (Menut et al., 2000) although perhaps it is antiquated.*

We have expanded the corresponding sentence accordingly. And indeed, an adjoint is not yet available for CHIMERE-2017.

Referee's comment: *Fig 4: Does the non-zero intercept of this line tell you something about the refractive index of the OC?*

A non–zero intercept ($\kappa_2$) is indeed indicative of an imaginary part of the refractive index of OC, but the interpretation of the intercept is not quite straightforward because the brown carbon content in aerosol particles can, in principle, correlate or anti-correlate with the BC content. A corresponding remark is introduced in Sect. 2.2.3 of the revised manuscript.

Referee's comment: *Fig 7(b): I'm not sure this shows much other than the fact that the modeled EC/(EC+OC) ratio is rather consistent, which is likely owing the fact that the model is based on a fixed EC/OC ratios in the emissions. Still, the authors should comment on the one feature that runs counter to this trend in the figure, the cluster of green points above the 1:1 line, which potentially indicates sampling of regions influenced by non-BB sources.*

Indeed, Fig. 7b indicates that the modeled EC/(EC+OC) ratio is rather consistent with the AAOD and AOD observations from the validation data subset. However, the EC/OC ratio is not fixed in the model but is proportional to the ratio of the correction factors which were optimized in our study.

In the revised manuscript, it is noted (see Sect. 3.2) that a cluster of green points above the regression line in Fig. 7b indicates a distinct contribution of agricultural and grass fires featuring much larger ratios of the BC and OC emission factors than the predominant forest fires.

Referee's comment: *23.7: The authors need to clarify the writing here with regards to precision vs accuracy. If the errors were truly random, they would not lead to bias.*

We meant to discuss the statistical significance of an estimate of the model bias. Of course, the true bias cannot be affected by random errors. To avoid the confusion, we have revised the corresponding sentence by using the word "difference" instead of "bias".

Referee's comment: *Fig 8: The way the background concentrations are included has*

*the downside of skewing the vertical scale such that differences between the model and observations appear minimized, visually. Instead, a linear scale should be used, and background concentrations could be plotted using a different vertical axis on the right hand side, or scaled by x10-100 so they could be plotted on the same scale. Also, more interesting than showing the background concentrations would be to show the model estimates when using the BB emissions with the optimized vs unoptimized emissions factors. Since "base" wasn't clearly defined in the presentation of the results, it's not clear to me which of these cases is being presented anyways.*

Following the Referee's suggestion, we have plotted the background concentrations by using a vertical axis on the right hand side of the figures. We attempted to use a linear scale instead of a logarithmic scale but found that many data points in May-June and August-September "collapsed" toward the zero, and so the difference between the simulated and observed values became indistinguishable. For this reason, we preferred to keep a logarithmic scale in Fig. 8. We also followed the Referee's suggestion to show the concentrations simulated using the unoptimized emission factors, even though we are not sure that such simulations are sufficiently meaningful (as noted above). Taking into account the magnitude of the optimized correction factors and a rather good agreement of the EC and OC concentrations observed at ZOTTO and simulated with the optimized emissions, it is not surprising that using the BB emission estimates calculated with the scaling factors of 1.0 resulted in an underestimation of both EC and OC concentrations.

Referee's comment: *Fig 10: The two cases don't appear to be significantly different; I would suggest just showing one. Also, I don't understand what the purple dots in panel (b) represent.*

Indeed, Fig. 10a and Fig. 10b do not appear to be significantly different, and this is actually an important result of our analysis of the YAK-AEROSIB observations. The matter of fact is that the number of the data points shown in Fig. 10b is two times larger than that in Fig. 10a (as indicated in the plots), but, in spite of such a major difference

between the corresponding datasets, the regression lines are almost identical. This result demonstrates the robustness of our conclusions based on the analysis of the aircraft measurements. Accordingly, we believe that showing the both figures makes our conclusions more convincing and, for this reason, we would prefer to keep both these panels in the paper. The meaning of the purple dots in Fig. 10b is explained in the figure caption: they mark the data points included in Fig. 10a. In the revised manuscript, we try to explain this more clearly. The size of the symbols and dots is enhanced to make them more visible.

Referee's comment: *24.27: I thought it was stated in the CHIMERE model description that there was no aging of BC included in the model.*

We write about the "representation of the BB aerosol aging processes in CHIMERE", meaning that these processes are mostly determined by the organic component of BB aerosol. There is no mentioning of aging of BC here.

Referee's comment: *26.7: While uncertainty information isn't provided with the GFED4 inventory, there have been numerous studies evaluating this inventory compared to other bottom-up inventories as well as evaluations based on top-down constraints using ambient measurements. These studies provide some range of estimates for the uncertainty in GFED4emissions that could be referenced. In other words, from previous work we know that it's not something small like 1%, and certainly error as large as x2 (or even x10, regionally) have been noted.*

We thank the Referee for this useful comment. Indeed, the available studies indicate that the GFED data may be rather uncertain. The discussion is expanded in the revised manuscript accordingly.

Referee's comments: *Editorial:*

*4.5 and other locations: Careful with random switches between present and past tense.*

*4.10: GEOS-Chem*

*8.12: "measurement data" is a little redundant.*

*10.32: distributions –> distribution*

*11.4: are anyway not –> are not*

*17.5: The authors cite Enting 2001, but the bibliography only includes Enting 2002.*

We thank the Referee for these editorial corrections. The corresponding changes are made in the revised manuscript.

Referee's comment: *Fig 2 and 9: Text in the figure is illegible. Suggest using solid black, font that is easier to read.*

Figures 2, 9 and also 5 are redrawn with more readable fonts.

Referee's comment: *Fig 5: Small suggestion on labeling: Don't write "CHIMERE - bgr" or "CHIMERE - base", as that at first looks like subtraction. Just leave out the hyphen, or use a colon.*

We thank you the Referee for this suggestion. We have redrawn the figures by using the underscore symbol in the simulation labels.

Referee's comment: *21.17: Figure 5?*

Yes, indeed. We are sorry for this misprint which is corrected in the revised manuscript.